# Proportional Participatory Budgeting
# with Additive Utilities

**Dominik Peters**
University of Toronto
Toronto, ON, Canada
dominik@cs.toronto.edu

**Grzegorz Pierczyński**
University of Warsaw
Warsaw, Poland
g.pierczynski@mimuw.edu.pl

**Piotr Skowron**
University of Warsaw
Warsaw, Poland
p.skowron@mimuw.edu.pl

## Abstract

We study voting rules for participatory budgeting, where a group of voters collectively decides which projects should be funded using a common budget. We allow the projects to have arbitrary costs, and the voters to have arbitrary additive valuations over the projects. We formulate two axioms that guarantee proportional representation to groups of voters with common interests. To the best of our knowledge, all known rules for participatory budgeting do not satisfy either of the two axioms; in addition we show that the most prominent proportional rule for committee elections, Proportional Approval Voting, cannot be adapted to arbitrary costs nor to additive valuations so that it would satisfy our axioms of proportionality. We construct a simple and attractive voting rule called the Method of Equal Shares that satisfies one of our axioms (for arbitrary costs and arbitrary additive valuations), and that can be evaluated in polynomial time. We prove that our other stronger axiom is also satisfiable, though by a computationally more expensive and less natural voting rule.

## 1 Introduction

Consider an abstract model where there is a group of agents who have preferences over a set of options. Each option has a cost, and the goal is to select a subset of options whose total cost does not exceed a predefined budget. This model provides a formal framework for a number of real-life scenarios. Perhaps the most natural example is Participatory Budgeting (PB). Through a voting system, PB allows residents of a city (the agents) to decide which projects (the options) will be funded by the government. In recent years, PB has been started in many cities around the world [Cabannes, 2004, Aziz and Shah, 2020], and in some cases is used to decide a significant fraction of the city budget. For example, in Paris, PB has been run every year since 2014, and since 2016 the total amount of funding for PB in Paris has been more than 100 million euros annually. Besides PB, the formal model captures the problem of electing a representative committee for a group of voters, say a faculty board or a parliament [Faliszewski et al., 2017, Lackner and Skowron, 2020], but also appears useful in situations that do not involve humans. For example, our framework describes the problem of selecting validators in consensus protocols, such as the blockchain [Cevallos and Stewart, 2020], the problem of selecting web pages that should be displayed in response to user queries, where the selected set of web pages should be useful for different types of user profiles [Skowron et al., 2017] or the problem of locating public facilities [Skowron et al., 2016, Byrka et al., 2018]. Algorithms for PB can even be used for improving the quality of genetic algorithms [Faliszewski et al., 2016]. While there are numerous applications of the model that we consider, for concreteness we will use terminology referring to participatory budgeting.

In this paper, we focus on designing algorithms (which we call aggregation rules) for selecting projects. We are interested in rules that are *fair* in the sense that each agent has roughly equal influence on the outcome. This implies that the selected subset of projects must proportionally represent the views of

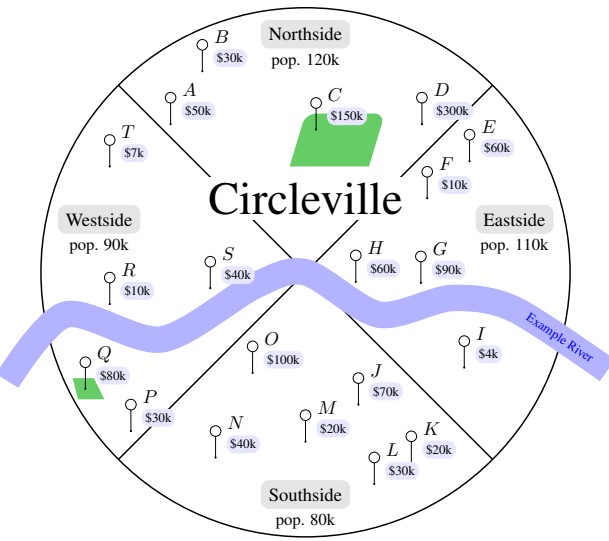

Figure 1: Map of Circleville, showing the locations and costs of the PB project proposals.

the voters, and so every group of voters with similar preferences should have an appropriate portion of the available budget allocated to fulfilling those preferences. Our results contribute to the study of fairness in algorithmic decision making. Notably, while many works on this topic aim to be fair to agents described by feature vectors (that for example contain demographic information), we only use agents' preferences. Fair algorithms will need to identify cohesive groups of agents on their own.

To understand the constraints that fairness and proportionality place on the decision procedure, let us start by discussing the way cities implement PB today. To count the votes, most cities use a variant of a simple protocol: Each voter is allowed to vote for a certain number of project proposals. Then, the projects with the highest number of votes are funded, until the budget limit is reached. While simple and intuitive, this is a bad voting rule. To see this, consider Circleville, a fictional city divided into four districts. A map of the city is shown in Figure 1. The districts all have similar sizes, but Northside has the largest population. Suppose $400k have been allocated to PB, and suppose that all the project proposals are of a local character (such as school renovations), and so residents only vote for projects that concern their own district. For example, every Northside resident will cast votes for projects $A$, $B$, $C$, and $D$, but no one else votes for these. Because Northside is the most populous district, the Northside projects will all receive the highest number of votes, and the voting rule described will spend the entire budget on Northside projects. The 280k residents of the other districts are left empty-handed.

To circumvent this obvious issue, many cities have opted to hold separate elections for each district. The budget is divided in advance between the districts (e.g., in proportion to their number of residents), each project is assigned to a district, and voters only vote in their local election. While this avoids the issue of spending the entire budget in Northside, this fix introduces many other problems. For example, projects on the boundary of two districts (such as $A$ and $P$) need to be assigned to one of them. Residents of the other district may be in favor of the boundary project, but cannot vote for it. Thus boundary projects are less likely to be funded, even if they would be more valuable overall. Similarly, projects without a specific location that benefit the entire city cannot be handled. Also, interest groups that are not geographic in nature will be underserved; for instance, parents across the city might favor construction of a large playground (project $C$), but with separate district elections, parents cannot form a voting block. Similarly, bike riders across the city cannot express their joint interest in the construction of a bike trail along Example River (projects $R$, $S$, $H$, and $G$).

To solve these problems, it seems desirable to hold a single city-wide election, but use a voting system that ensures that money is spent proportionally. The voting system should automatically and endogenously identify groups of voters who share common interests, and make sure that those groups are appropriately represented. This aim has been identified by several researchers [Aziz et al., 2018b], but no convincing proposal for a proportional voting rule has emerged so far. Indeed, no good formalization of "proportionality" for the PB context has been identified in the literature, except for

the concept of the *core*. However, the core is a very demanding requirement, and there are situations where it fails to exist [Fain et al., 2018].

In this paper, we formalize proportionality for participatory budgeting as an axiom called *extended justified representation* (EJR). The axiom requires that no group of voters with common interests is underserved. We construct a simple and attractive voting rule (the *Method of Equal Shares*) that satisfies EJR for approval preferences, and that satisfies EJR up to one project for general additive valuations. We then discuss a strengthening of EJR—which we call *fully justified representation* (FJR)—and show that this strengthening is still satisfiable, albeit by a different voting rule.

Both our proportionality axiom and our voting rule are generalizations of concepts that have been introduced in the literature on multi-winner voting [Faliszewski et al., 2017]. That literature can be seen as handling a special case of PB, where all projects cost the same amount of money. This is often called the *unit cost assumption*. It turns out that the unit cost assumption substantially simplifies the problem. Further, much of the relevant literature studies rules that work with *approval ballots*, where voters are allowed to approve or disapprove each project. In our paper, we allow any additive valuations (not just 0/1), which is more expressive. The proportionality axioms and voting rules that we introduce all work for general additive valuations. This is notable, since allowing additive valuations introduces significant conceptual difficulty. Indeed, most prominent multi-winner voting rules do not naturally extend to additive valuations (or at least not gracefully).

Allowing additive valuations gives voters a way to more precisely describe their preference intensities. This can be valuable information in a PB setting where there are often projects which differ significantly in their cost. In practice, PB elections typically only elicit approval information. Using rules such as ours (defined for general additive valuations), we can interpret this approval information in two different ways: either as 0/1 utilities (so that a voter's utility for a selected outcome is the number of approved funded projects) or as cost-based utilities (so the utility for an outcome is the total cost of approved funded projects). These two interpretations lead to interestingly different rules, with cost-based utilities favoring more expensive projects and leading to outcomes that are more similar to the outcomes that are selected by the greedy rule typically used by cities today.

## 2 Preliminaries

For each $t \in \mathbb{N}$, write $[t] = \{1, 2, \ldots, t\}$. An *election* is a tuple $(N, C, \text{cost}, \{u_i\}_{i \in N})$, where:

- $N = [n]$ and $C = \{c_1, \ldots, c_m\}$ are the sets of *voters* and *candidates* (or *projects*).
- $\text{cost} \colon C \to \mathbb{Q}_+$ is a function that for each $c \in C$ assigns the *cost* that needs to be paid if $c$ is selected. For each $T \subseteq C$, we write $\text{cost}(T) = \sum_{c \in T} \text{cost}(c)$ for the total cost of $T$.
- For each voter $i \in N$, the function $u_i \colon C \to [0, 1]$ defines $i$'s additive *utility function*. If a set $T \subseteq C$ of candidates is implemented, $i$'s overall utility is $u_i(T) = \sum_{c \in T} u_i(c)$. For a subset $S \subseteq N$ of voters, we further write $u_S(T) = \sum_{i \in S} \sum_{c \in T} u_i(c)$ for the total utility enjoyed by $S$ if $T$ is implemented. We assume that $u_N(c) > 0$ for each $c \in C$, so that every candidate is assigned positive utility by at least one voter.

The voters have a fixed common *budget* which we normalize to 1. A subset of candidates $W \subseteq C$ is *feasible* if $\text{cost}(W) \leq 1$. Our goal is to choose a feasible subset of candidates, which we call an *outcome*, based on voters' utilities. An *aggregation rule* (or, in short, a *rule*) is a function $\mathcal{R}$ that for each election $E$ returns a feasible outcome $W = \mathcal{R}(E)$ called the *winning outcome*.[1]

There are two interesting special cases of our model:

**Committee elections.** In this case, there exists $k \in \mathbb{N}$ (the committee size) such that each candidate costs $1/k$. Then $W$ is an outcome if and only if $|W| \leq k$. In this special case we also refer to outcomes as *committees*, and we say that the election satisfies the *unit cost assumption*.

**Approval utilities.** In this case, for each $i \in N$ and $c \in C$ it holds that $u_i(c) \in \{0, 1\}$. The *approval set* of voter $i$ is $A(i) := \{c \in C \colon u_i(c) = 1\}$, and we say that $i$ *approves* candidate $c$ if $c \in A(i)$. If $c \in A(i) \cap W$, we say that $c$ is a *representative* of $i$.

Often we combine of these special cases, and study approval-based committee elections.

---

[1]Sometimes there are ties. For the results of this paper it does not matter how these ties are broken.

# 3 The Method of Equal Shares (MES)

Recently, Peters and Skowron [2020] introduced an aggregation rule for approval-based committee elections that they called Rule X. In that setting the rule satisfies a combination of appealing proportionality properties. Here, we extend it to the more general model of participatory budgeting, that is, to the model with arbitrary costs and utilities. We will call this rule the Method of Equal Shares (in short, MES).

**Definition 1** (Method of Equal Shares (MES)). Each voter is initially given an equal fraction of the budget, i.e., each voter is given $1/n$ dollars. We start with an empty outcome $W = \emptyset$ and sequentially add candidates to $W$. To add a candidate $c$ to $W$, we need the voters to pay for $c$. Write $p_i(c)$ for the amount that voter $i$ pays for $c$; we will need that $\sum_{i \in N} p_i(c) = \text{cost}(c)$. We write $p_i(W) = \sum_{c \in W} p_i(c) \leq \frac{1}{n}$ for the total amount $i$ has paid so far. For $\rho \geq 0$, we say that a candidate $c \notin W$ is $\rho$-affordable if

$$\sum_{i \in N} \min\left(\tfrac{1}{n} - p_i(W), u_i(c) \cdot \rho\right) = \text{cost}(c).$$

If no candidate is $\rho$-affordable for any $\rho$, MES terminates and returns $W$. Otherwise it selects a candidate $c \notin W$ that is $\rho$-affordable for a minimum $\rho$. Individual payments are given by

$$p_i(c) = \min\left(\tfrac{1}{n} - p_i(W), u_i(c) \cdot \rho\right) \qquad \qquad \square$$

Intuitively, when the Method of Equal Shares (MES) adds a candidate $c$, it asks voters to pay an amount proportional to their utility $u_i(c)$ for $c$; in particular, the cost per unit of utility is $\rho$. If a voter does not have enough money, the rule asks the voter to pay all the money the voter has left, which is $\frac{1}{n} - p_i(W)$. Throughout the execution of MES, the value of $\rho$ increases. Thus, candidates are added in decreasing order of utility per dollar that the voters get from the candidates. In comparison to the work of Peters and Skowron [2020], the new elements in our definition of the rule are (1) the formula according to which the costs of selected projects are divided among the voters, and (2) the algorithm specifying in which order the candidates are selected; these are critical to ensure that the rule is proportional.

## 3.1 Extended Justified Representation (EJR)

The first notion of proportionality that we examine is Extended Justified Representation (EJR). This axiom was first proposed for approval-based committee elections [Aziz et al., 2017]. Even for the special case of approval-based committee elections, only few rules are known to satisfy EJR [Aziz et al., 2017, 2018a, Peters and Skowron, 2020], but the Method of Equal Shares is one of them. In this section, we introduce a generalization of EJR to the PB model and show that our rule continues to satisfy EJR.

We first recall the definition of EJR for approval-based committee elections. Intuitively, this axiom ensures that every large enough group of voters whose approval sets have a large enough intersection must obtain a fair number of representatives. For example, if a group of voters forms an $\alpha$-fraction of the whole population and if this group agrees on sufficiently many candidates, then it should be allowed to decide about an $\alpha$-fraction of the elected candidates. Formally, this is achieved by excluding the possibility that each member of the group approves less than $\lfloor \alpha k \rfloor$ elected candidates.

**Definition 2** (Extended Justified Representation for approval-based committee elections). We say that a group of voters $S$ is $\ell$-cohesive for $\ell \in \mathbb{N}$ if $|S| \geq \ell/k \cdot n$ and $|\bigcap_{i \in S} A(i)| \geq \ell$. A rule $\mathcal{R}$ satisfies *extended justified representation* if for each election instance $E$ and each $\ell$-cohesive group $S$ of voters there exists a voter $i \in S$ such that $|A(i) \cap \mathcal{R}(E)| \geq \ell$.

At first sight it is unintuitive that we only require that at least one voter obtain $\ell$ representatives. However, the strengthening of EJR that requires each member of $S$ to obtain $\ell$ representatives is impossible even on very small instances [Aziz et al., 2017]. Still, even with only the at-least-one guarantee, EJR has plenty of bite [Aziz et al., 2018a, Skowron, 2018, Peters and Skowron, 2020].

The generalization of this axiom to the PB model is not straightforward and to the best of our knowledge none has been proposed in the literature.[2] To warm up, let's first relax the unit cost assumption, but stay in the approval-based setting. Then EJR should state the following.

**Definition 3** (Extended Justified Representation for approval-based elections). We say that a group of voters $S$ is $T$-cohesive for $T \subseteq C$ if $|S| \geq \text{cost}(T) \cdot n$ and $T \subseteq \bigcap_{i \in S} A(i)$. A rule $\mathcal{R}$ satisfies *extended justified representation* if for each election instance $E$ and each $T$-cohesive group $S$ of voters there exists a voter $i \in S$ such that $|A(i) \cap \mathcal{R}(E)| \geq |T|$.

Thus, cohesiveness now requires that the group $S$ can identify a collection of projects $T$ that they all approve and that is affordable with their fraction of the budget ($|S| \geq \text{cost}(T) \cdot n$). Note that voters $i \in S$ obtain utility $u_i(T) = |T|$ from $T$; EJR requires that at least one member of $S$ must attain this utility in the election outcome.

To further generalize EJR beyond approvals is more difficult, because the notion of a candidate who is approved by all members of $S$ does not have an analogue. Instead, we quantify cohesion by calculating the minimum utility that any member of $S$ assigns to each project in $T$.

**Definition 4** (Extended Justified Representation). A group of voters $S$ is $(\alpha, T)$-*cohesive*, where $\alpha \colon C \to [0; 1]$ and $T \subseteq C$, if $|S| \geq \text{cost}(T) \cdot n$ and if $u_i(c) \geq \alpha(c)$ for all $i \in S$ and $c \in T$. A rule $\mathcal{R}$ satisfies *extended justified representation* if for each election instance $E$ and each $(\alpha, T)$-cohesive group of voters $S$ there exists a voter $i \in S$ such that $u_i(\mathcal{R}(E)) \geq \sum_{c \in T} \alpha(c)$.

Again, an $(\alpha, T)$-cohesive group of voters $S$ can propose the projects in $T$, since they are affordable with $S$'s share of the budget. The values $(\alpha(c))_{c \in T}$ denote how much the coalition $S$ agrees about the desirability of the projects in $T$. In particular, we have $u_i(T) \geq \sum_{c \in T} \alpha(c)$ for each $i \in S$. Consequently, Definition 4 prohibits any outcome in which every voter in $S$ gets utility strictly lower than $\sum_{c \in T} \alpha(c)$; hence there must exists $i \in S$ such that $u_i(\mathcal{R}(E)) \geq \sum_{c \in T} \alpha(c)$.

EJR is a demanding property in the PB model. Consider the special case where there is only one voter, $N = \{1\}$. Then any outcome $W$ satisfying EJR must solve the knapsack problem, i.e. it must maximize $\sum_{c \in W} u_1(c)$, since otherwise an optimum knapsack $T$ witnesses an EJR violation. Because the knapsack problem is weakly NP-hard, this presents a difficulty for a rule to satisfy EJR.[3]

**Proposition 1.** *Unless P = NP, no aggregation rule that can be computed in strongly polynomial time can satisfy EJR in the general PB model.*

Indeed, the Method of Equal Shares fails EJR in the general PB model. However, we can show that it satisfies a mild relaxation, which requires EJR to hold "up to one project".

**Definition 5** (Extended Justified Representation Up To One Project). A rule $\mathcal{R}$ satisfies extended justified representation *up to one project* if for each election instance $E$ and each $(\alpha, T)$-cohesive group of voters $S$ there exists a voter $i \in S$ such that either $u_i(\mathcal{R}(E)) \geq \sum_{c \in T} \alpha(c)$ or for some $a \in C$ it holds that $u_i(\mathcal{R}(E) \cup \{a\}) > \sum_{c \in T} \alpha(c)$.

It is worth noting that in the approval-based model, Definitions 4 and 5 are actually equivalent, because the "up to one project" option never applies: Consider an $(\alpha, T)$-cohesive group of voters $S$. Since voters' utilities are 0/1, we may assume that for each $c \in T$ we have $\alpha(c) = 1$: if $\alpha(c) > 0$ this is clear; otherwise we can remove $c$ from $T$ without losing cohesiveness. Thus, the cohesiveness condition is equivalent to the condition that every voter approves every candidate in $T$. Finally, note that in the approval model, due to the strict inequality, both conditions $u_i(\mathcal{R}(E)) \geq \sum_{c \in T} \alpha(c)$ and $\exists_{a \in C}.u_i(\mathcal{R}(E) \cup \{a\}) > \sum_{c \in T} \alpha(c)$ boil down to $|A(i) \cap \mathcal{R}(E)| \geq \sum_{c \in T} \alpha(c) = |T|$.

Our main result is that the Method of Equal Shares satisfies EJR up to one project in the general PB model. By the previous observation, it hence satisfies EJR in the approval-based model (even when not imposing unit costs).

**Theorem 1.** *The Method of Equal Shares satisfies EJR up to one project in the participatory budgeting model.*

---

[2] Aziz et al. [2018b] generalize the weaker axiom of Proportional Justified Representation (PJR) [Sánchez-Fernández et al., 2017] beyond unit costs, but they operate in a non-standard utility model where voters care more about more expensive projects.

[3] Aziz et al. [2018b, Prop. 3.8] prove a similar result for their BPJR-L notion, by reduction from subset sum.

*Proof.* For a contradiction, assume there is an election $E$, a set $S \subseteq N$ and a set $T \subseteq C$ such that: (i) $\text{cost}(T) \leq {}^{|S|}/{}_n$, (ii) $u_i(c) \geq \alpha(c) > 0$ (candidates with $\alpha(c) = 0$ can be skipped) for each $i \in S$ and $c \in T$, and (iii) $u_i(\mathcal{R}(E) \cup \{a\}) \leq \sum_{c \in T} \alpha(c)$ for each $i \in S$ and $a \in T$.

Assume for a while that the voters from $S$ have unrestricted initial budgets, and let us analyze how MES would proceed in this case. For simplicity, let us rename the candidates in $T$ so that $T = \{c_1, \ldots, c_t\}$ and so that for $1 \leq i < j \leq t$ candidate $c_i$ is picked by MES before candidate $c_j$.

Whenever a candidate $c \in T$ is selected, the voters pay for this candidate. Voter $i$ pays $p_i(c)$ dollars for $c$, and in return, she gets utility $u_i(c)$. Thus, the price-per-utility she pays equals $\rho_i(c) = {}^{p_i(c)}/{}_{u_i(c)}$. MES works in a way that all voters from $S$ who pay for $c$ obtain the same price-per-utility ratio, i.e., for all $i, j \in S$ and $c \in C$ we have that $\rho_i(c) = \rho_j(c)$. Further, this price-per-utility equals at most ${}^{\text{cost}(c)}/{}_{u_S(c)}$, independently of whether any voters from $N \setminus S$ pay for $c$ or not (if no voters from $N \setminus S$ pays for $c$, then the price-per-utility equals exactly ${}^{\text{cost}(c)}/{}_{u_S(c)}$):

$$\rho_i(c) = \frac{p_i(c)}{u_i(c)} = \frac{p_i(c) \cdot \frac{\sum_{j \in S} u_j(c)}{\sum_{j \in S} u_j(c)}}{u_i(c)} = \frac{1}{\sum_{j \in S} u_j(c)} \cdot \sum_{j \in S} \frac{p_i(c)}{u_i(c)} \cdot u_j(c) = \frac{\sum_{j \in S} p_j(c)}{\sum_{j \in S} u_j(c)} \leq \frac{\text{cost}(c)}{u_S(c)}.$$

Since $u_i(c) \geq \alpha(c)$ for each $i \in S$ and $c \in T$, the price-per-utility for $c \in T$ equals at most ${}^{\text{cost}(c)}/{}_{|S|\alpha(c)}$. Now, consider the voter who in the first possible iteration uses more than ${}^1/{}_n$ dollars[4]. For this voter, call her $i$, let us consider the function $f$ defined as follows. For each value $x$, the function $f$ returns the price that $i$ needs to pay to achieve the utility $x$. We make this function continuous, by assuming that the candidates are divisible. That is, if the voter pays $p$ for her first paid candidate $c$ with utility $u_i(c)$, then $f({}^{u_i(c)}/{}_2) = {}^p/{}_2$, $f({}^{u_i(c)}/{}_3) = {}^p/{}_3$, and so on. The key observation is that the function $f$ is convex. This is because MES selects the candidates in increasing order of price-per-utility. This function is depicted below.

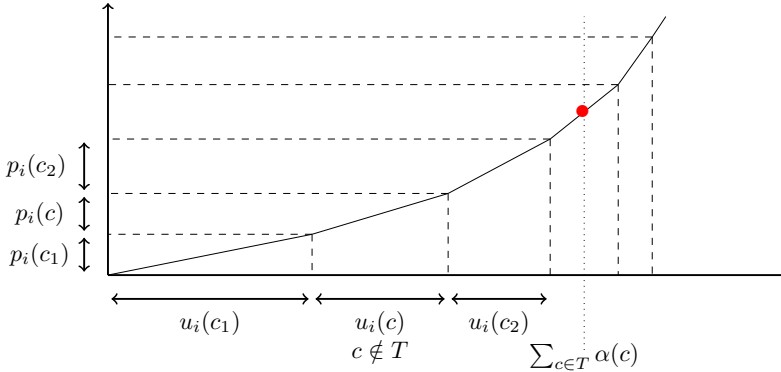

We are interested in the value $f(\sum_{c \in T} \alpha(c))$. This value would be maximized if the fragments of the function with the lowest slope were the shortest. However, we know that the part of the function that corresponds $\rho_i(c)$ must be of length at least equal to $u_i(c) \geq \alpha(c)$. Thus:

$$f\left(\sum_{c \in T} \alpha(c)\right) \leq \sum_{c \in T} \alpha(c) \cdot \rho_i(c) \leq \sum_{c \in T} \alpha(c) \cdot \frac{\text{cost}(c)}{|S|\alpha(c)} = \sum_{c \in T} \frac{\text{cost}(c)}{|S|} = \frac{\text{cost}(T)}{|S|} \leq \frac{1}{n}.$$

Now, consider the first moment when $i$ uses more than ${}^1/{}_n$ dollars. Until this time moment, MES behaves exactly in the same way as if the voters from $S$ had their initial budgets set to ${}^1/{}_n$ (this follows from how we chose $i$). Further, we know that in this moment, if we chose a candidate that would be chosen if the voters had unrestricted budgets, then the utility of voter $i$ would be greater than $\sum_{c \in T} \alpha(c)$. This gives a contradiction, and completes the proof. $\square$

Theorem 1 establishes the Method of Equal Shares as a prime candidate for voting under a budget constraint, showing that it satisfies a demanding fairness property. This makes it the first known rule

---

[4]The only case when there is no such voter is when every candidate $c \in C$ with $u_S(c) > 0$ has already been elected before. But then the utility of $i$ from the elected outcome is clearly at least $\sum_{c \in T} u_i(c) \geq \sum_{c \in T} \alpha(c)$.

that can give such strong proportionality guarantees in the model with additive utilities and arbitrary costs. In the literature on the special case of approval-based committee elections, another rule has received much attention: Proportional Approval Voting (PAV). Let us briefly recall the definition of this rule.

**Definition 6** (Proportional Approval Voting (PAV)). For an approval-based election, PAV selects a feasible outcome maximizing $\sum_{i \in N} H(|A(i) \cap W|)$, where $H(r) = \sum_{j=1}^{r} 1/j$ is the $r$th harmonic number.

This rule satisfies EJR when assuming unit costs [Aziz et al., 2017]. But without unit costs, PAV fails EJR. In fact, Example 1 shows that, for each $r \geq 0$, PAV does not even satisfy EJR up to $r$ projects.

**Example 1.** Fix a constant $r \in \mathbb{N}$ $(r \geq 2)$, and consider the following approval-based profile:

$$
\begin{aligned}
r^2 - 1 \text{ voters:} &\qquad \{a_1, a_2, \ldots, a_r\}, \\
1 \text{ voter:} &\qquad \{b_1, b_2, \ldots, b_r\}.
\end{aligned}
$$

The candidates $a_1, a_2, \ldots, a_r$ cost $1/r$ dollars each; the candidates $b_1, b_2, \ldots, b_r$ cost $1/r^3$ dollars each. EJR requires that the one voter who approves candidates $b_1, \ldots, b_r$ must approve at least $r$ candidates in the outcome. However, PAV selects $\{a_1, a_2, \ldots, a_r\}$, leaving the voter with nothing. $\square$

In fact, in Appendix A, we prove that no rule that globally maximizes an objective function over voter utilities (like PAV) can satisfy proportionality. Further, in Appendix C we argue that another rule for proportional approval-based committee elections, Phragmén's rule, does not extend to the PB setting.

## 3.2 Approximating the Core

An important proportionality property that has been proposed for PB is the *core* [Aziz et al., 2017, Fain et al., 2018], an idea adapted from cooperative game theory.

**Definition 7** (The Core). For an election $(N, C, \text{cost}, \{u_i\}_{i \in N})$, an outcome $W$ is in the *core* if for every $S \subseteq N$ and $T \subseteq C$ with $|S| \geq \text{cost}(T) \cdot n$ there exists $i \in S$ such that $u_i(W) \geq u_i(T)$.

The core is a stronger guarantee than EJR. The core allows any group $S$ to present an arbitrary "counter-proposal" $T$ that they can afford, and guarantees that at least one member $i \in S$ would prefer to stick with the core outcome $W$, so $u_i(W) \geq u_i(T)$. EJR only guarantees that $u_i(W) \geq \sum_{c \in T} \min_{j \in S} u_j(c)$. Thus, EJR only respects counter-proposals $T$ if they have broad agreement within the coalition $S$. This is arguably a reasonable restriction, since such coalitions can more easily coordinate to "complain" against the selected $W$. Still, it would be nice to give the stronger core guarantee. Unfortunately, there are elections where no outcome is in the core, even with unit costs.

**Example 2.**[5] We have 6 voters and 6 candidates with unit costs, and $k = 3$. Utilities satisfy

$$
\begin{aligned}
u_1(c_1) > u_1(c_2) > 0, &\qquad u_2(c_2) > u_2(c_3) > 0, &\qquad u_3(c_3) > u_3(c_1) > 0; \\
u_4(c_4) > u_4(c_5) > 0, &\qquad u_5(c_5) > u_5(c_6) > 0, &\qquad u_6(c_6) > u_6(c_4) > 0,
\end{aligned}
$$

and all other utilities are equal to 0. Let $W \subseteq C$ be any feasible outcome, so $|W| \leq 3$. Then either $|W \cap \{c_1, c_2, c_3\}| \leq 1$ or $|W \cap \{c_4, c_5, c_6\}| \leq 1$. Without loss of generality assume the former, and assume that $c_2 \notin W$ and $c_3 \notin W$. Then $S = \{v_2, v_3\}$ and $T = \{c_3\}$ block $W$, since $2 = |S| \geq \text{cost}(T) * n = \frac{1}{3} \cdot 6 = 2$ and both $u_2(c_3) > u_2(c_1) \geq u_2(W)$ and $u_3(c_3) > u_3(c_1) \geq u_3(W)$. $\square$

Notably, this example is not approval-based. It is unknown whether the core is always non-empty for approval-based elections (with or without the unit cost assumption).

In the committee context, Peters and Skowron [2020] showed that the Method of Equal Shares (MES) returns an outcome that never violates the core too badly. We can generalize this result to the general PB setting: MES provides a multiplicative approximation to the core.[6]

**Definition 8.** For $\alpha \geq 1$, we say that an outcome is in the *$\alpha$-core* if for every $S \subseteq N$ and $T \subseteq C$ with $|S| \geq \text{cost}(T) \cdot n$ there exists $i \in S$ and $c \in T$ such that $u_i(\mathcal{R}(E) \cup \{c\}) \geq \frac{u_i(T)}{\alpha}$.

---

[5] This example is adapted from Fain et al. [2018, Appendix C] so as to satisfy the unit cost assumption.

[6] Our approximation notion differs from one proposed by Fain et al. [2018] (which also involves an additive term) and one proposed by Cheng et al. [2019] and Jiang et al. [2020] which approximates the coalition size.

**Theorem 2.** *Given an election $E$, let $u_{\max}$ be the highest utility a voter can get from a feasible outcome. Let $u_{\min}$ we denote the smallest, yet positive utility a voter can get from a feasible outcome:*

$$u_{\max} = \max_{i \in N} \max_{\text{cost}(W) \leq 1} u_i(W) \qquad and \qquad u_{\min} = \min_{i \in N} \min_{u_i(W) > 0} u_i(W).$$

*Then the outcome selected by MES is always in the $\alpha$-core for $\alpha = 4 \log(2 \cdot u_{\max}/u_{\min})$.*

The bound of $\alpha$ is asymptotically tight. All proofs omitted from the main text appear in Appendix D.

### 3.3 Other properties of the Method of Equal Shares

In Appendices B and C we discuss two other properties: priceability [Peters and Skowron, 2020, Peters et al., 2021] and exhaustiveness. Here, let us only discuss the latter one. Exhaustiveness requires that a voting rule spends its entire budget. Of course, due to the discrete model, we cannot guarantee that the rule will spend exactly 1 dollar (i.e., the entire budget); however, we can require that no additional project is fits within the budget.

**Definition 9** (Exhaustiveness, Aziz et al., 2018b). An election rule $\mathcal{R}$ is *exhaustive* if for each election instance $E$ and each non-selected candidate $c \notin \mathcal{R}(E)$ it holds that $\text{cost}(\mathcal{R}(E) \cup \{c\}) > 1$.

Notably, the Method of Equal Shares fails to be exhaustive: even if there is enough budget remaining to fund more projects, the rule may reach a state when no project is $\rho$-affordable. In Appendix C we discuss a few possible ways to make the rule exhaustive. We experimentally compare these modifications in Appendix H.

## 4 Greedy Cohesive Rule

In Section 3 we discussed the EJR axiom for the PB model, and saw that it is implemented by Method of Equal Shares. We will now propose a strengthening of EJR, and show a rule that satisfies the new strong property. Interestingly, even in the approval-based committee-election model our new property is substantially stronger than EJR, and hence this new rule provides the strongest known proportionality guarantees. On the other hand, compared to MES, it is computationally expensive and arguably less natural.

### 4.1 Full Justified Representation (FJR)

Our new proportionality axiom strengthens EJR by weakening its requirement that groups must be cohesive. Thus, the new axiom guarantees representation to groups that are only partially cohesive.

**Definition 10** (Full Justified Representation (FJR)). We say that a group of voters $S$ is *weakly $(\beta, T)$-cohesive* for $\beta \in \mathbb{R}$ and $T \subseteq C$, if $|S| \geq \text{cost}(T) \cdot n$ and $u_i(T) \geq \beta$ for every voter $i \in S$. A rule $\mathcal{R}$ satisfies *full justified representation (FJR)* if for each election instance $E$ and each weakly $(\beta, T)$-cohesive group of voters $S$ there exists a voter $i \in S$ such that $u_i(\mathcal{R}(E)) \geq \beta$.

In the approval-based committee-election model, FJR boils down to the following requirement: Let $S$ be a group of voters, and suppose that each member of $S$ approves at least $\beta$ candidates from some set $T \subseteq C$ with $|T| \leq \ell$, and let $|S| \geq \ell/k \cdot n$. Then at least one voter from $S$ must have at least $\beta$ representatives in the committee. It is clear that in the special case of $\beta = \ell$, we obtain Definition 2, hence FJR implies EJR. The same implication holds in the general PB model.

**Proposition 2.** *FJR implies EJR in the general PB model.*

It is easy to see that FJR is implied by the core property (cf. Definition 7). It is related to, but stronger than, some other relaxations of the core discussed by Peters and Skowron [2020, Section 5.2].

Previously known aggregation rules fail FJR (see Appendix E). Still, it turns out that FJR can always be satisfied: we present a (somewhat artificial) rule satisfying this strong notion of proportionality.

**Definition 11** (Greedy Cohesive Rule (GCR)). The *Greedy Cohesive Rule* (GCR) is defined sequentially as follows: we start with an empty outcome $W = \emptyset$. At each step, we search for a weakly $(\beta, T)$-cohesive group $S$. If such a group exists, we find one where $\beta \geq 1$ is maximum,[7] add all the

---

[7]In Appendix F.1 we study additional properties of GCR. For those results it is useful, at this stage, to break ties in favor of smaller $\text{cost}(T)$. The proof of Theorem 3 does not depend on the way we break ties.

candidates from $T$ to $W$, remove all voters in $S$ from the election and repeat the search. If no such group exists, we stop and return $W$.

Let us first check that the Greedy Cohesive Rule always selects an outcome that does not exceed the budget limit. Indeed, whenever the algorithm adds some set $T$ to $W$, then by definition of weakly cohesive groups, we have $|S| \geq \text{cost}(T) \cdot n$, and hence it removes at least $\text{cost}(T) \cdot n$ voters after this step. Thus, if GCR selects an outcome with total cost $\text{cost}(W)$, then it must have removed at least $\text{cost}(W) \cdot n$ voters during its execution. Hence $\text{cost}(W) \leq 1$.

**Theorem 3.** *The Greedy Cohesive Rule satisfies FJR.*

In Appendix F we further analyze GCR and discuss ways to extend its outcome when it is not exhaustive. We also present an example where GCR selects a counter-intuitive outcome.

## 5 Experiments

In this section we evaluate different voting rules on data from real-world participatory budgeting elections carried out in several major cities in Poland. The data we use is taken from Pabulib and was collected by [Stolicki et al., 2020].[8] The data is anonymized except for some basic demographic information which we do not use in our experiments.

We looked at election instances in which the projects were divided into several groups. One group consists of city-wide projects, and each other group consists of projects that were assigned to a city district. Each voter was allowed to approve at most ten city-wide projects, and at most ten projects from her district. A part of the municipal budget was assigned to city-wide projects and the other part was divided among the districts in proportion to their populations. Currently, the cities that we consider use a rule that selects projects greedily in order of approval score until the budget is exhausted.

In our experimental analysis we used two types of voters' preferences:

**Approval utilities:** corresponding directly to the approval-ballots from our PB data.

**Cardinal utilities:** for each voter $i$ and each project $c_j$ we obtained the utility $u_i(c_j)$ as follows. If $i$ does not approve $c_j$, we set $u_i(c_j) = 0$. If $i$ approves $c_j$, we sample $u_i(c_j)$ from the normal distribution centred at $\text{cost}(c_j)$. (We also tested similar models where $u_i(c_j)$ was sampled from the uniform and exponential distributions, but those led to qualitatively similar conclusions.)

We are interested in comparing the Method of Equal Shares, the greedy approval rule currently used for selecting projects, Phragmén's rule, and the sequential version of PAV (sPAV). (We have limited the experiments to polynomial-time algorithms, since the instances are quite large.) Since Phragmén's rule does not extend to additive utilities, we only use this rule for approval utilities.

In our analysis we evaluated the following metrics:

**Total utility (UTIL).** The total utility of the voters from the selected set $W$: $\sum_{i \in N} \sum_{c \in W} u_i(c)$.

**Distribution of projects (PROJ-DIS).** For each election instance we look at the projects selected from each district. We compute their cost and divide it by the fraction of the budget that is proportional to the population of the district. From those ratios we take a variance.

**Distribution of utilities (UTIL-DIS).** For each election instance and each voter $i$ we compute her normalised utility from the set of selected projects $W$, which we define as $\sum_{i \in N} \sum_{c \in W} u_i(c)$ divided by $n \cdot \sum_{c \in W} u_i(c)$. We compute the variance of these values.

In Appendix H.1 we discuss in detail the obtained results and provide tables summarizing the measured metrics. Below we only briefly discuss the conclusions from the experimental evaluation.

First we checked whether the outcome that was in fact selected by the cities is fair according to the kind of fairness criteria we have been studying. We found that in at least 59 out of 366 elections (16%), EJR was failed. In most cases, the failure was of the form that there was a group of voters who approved 0 of the selected projects but who approved an unelected project in common, and the

---

[8]The data is publically available at pabulib.org.

group was large enough to afford that project. This is even a failure of the PB version of the axiom JR (Justified Representation).

Second, we compared three different strategies of making MES exhaustive. We observed substantial differences between different variants of MES. We conclude that MES gives a lot of flexibility to a mechanism designer, as it often selects outcomes that do not spend all of the budget, while still satisfying strong fairness requirements like EJR. Depending on the specific objectives, a mechanism designer can choose to complete this outcome using different strategies. Among the strategies we described in Appendix C, we observed that the outcomes produced by EXH2 are better from a utilitarian perspective. It also divides the budget between different city districts in a substantially fairer way than outcomes produced by EXH1. Therefore we suggest EXH2 as the preferred method.

In our third experiment we compared MES, Phragmén's rule, and PAV. We observed that for approval utilities the results returned by MES and Phragmén's rule are comparably good, both in terms of the total utility obtained by the voters and in terms of proportionality measured as a distribution of projects and voters' utilities. On the other hand, if we take a model with more fine-grained utilities, the difference between the two rules becomes apparent. This difference is unsurprising since Phragmén's rule does not take into account the more fine-grained information on utilities, but operates only on approval ballots. Yet, our results suggest that there is indeed a considerable advantage of using rules (like MES) that take into account the full information contained in cardinal additive utilities. We conclude that MES performs as well as Phragmén's rule for approval ballots and outperforms it when more fine-grained information on voters' utilities are available. Somewhat surprisingly, we show that the sequential variant of PAV produces highly disproportional outcomes compared to Phragmén's rule and MES. Altogether, our experiments confirm our theoretical results and suggest that the Method of Equal Shares outperforms the other two rules in terms of proportionality and/or efficiency.

## 6 Conclusion

In this paper, we have formulated two axioms, EJR and FJR, that capture the idea of proportionality in the participatory budgeting (PB) model. We have argued that none of the prominent committee election rules extend to the PB model so that it would satisfy even much weaker forms of proportionality. We have designed a simple and natural rule for the PB model, the Method of Equal Shares (MES). It satisfies EJR and other proportionality-related properties, and it is computable in polynomial time. The stronger of our two properties, FJR, is also satisfiable, albeit by a different and arguably less natural voting rule. It is an interesting open question whether there exists a natural voting rule that satisfies FJR and shares other desirable properties of MES.

There are numerous results which are not included in the main text. Specifically, in Appendices B and C we discuss other properties of MES, and in Appendix F we provide a more detailed analysis of GCR. In Appendix G we explain that our rules can be directly applied to a model where voters have ordinal preferences, that is, when they rank the candidates from the most to the least preferred one. Notably, in the ordinal model MES satisfies the axiom of proportionality for solid coalitions, which is perhaps the strongest known axiom of proportionality for ordinal preferences.

Many cities run PB by dividing the overall budget between districts and running separate elections in each district. In particular, this is true in the Polish cities that provide our experimental election data. We claimed in the introduction that this practice of separate elections leads to inferior outcomes. We designed a final experiment to study this question. Our results show a visible advantage of using global rules such as MES over separate district elections. For example, MES always produces outcomes with a more equal distribution of voter utility, and in most cases also provides a higher total utility in comparison to the rules that are in actual use in the elections we examined.

### Acknowledgments

Grzegorz Pierczyński and Piotr Skowron were supported by Poland's National Science Center grant UMO-2019/35/B/ST6/02215.

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
