 

Figure 2: Onetown and Twotown are identical, except that the projects have different costs. Both have a budget of $90k available for PB.

# A  Proportional Approval Voting is not proportional, nor is any variant of it

Probably the most popular multi-winner voting rule is Proportional Approval Voting (PAV), also known as Thiele's method after its Danish first inventor Thiele [1895]. Thiele's rule is based on optimization. Suppose that $N$ is the set of voters, and that each voter $i \in N$ has indicated a set $A(i)$ of projects that $i$ approves. Then for each set $W$ of projects which is feasible (i.e. its total cost is at most the budget limit), the rule computes the score

$$\text{PAV-score}(W) = \sum_{i \in N} \left( 1 + \frac{1}{2} + \frac{1}{3} + \cdots + \frac{1}{|W \cap A(i)|} \right).$$

The output of PAV is a feasible set $W$ that maximizes this score. When the unit cost assumption holds, PAV is a great rule and lives up to its name: it is known to be proportional both in an axiomatic sense [Aziz et al., 2017] and in a quantitative sense [Skowron, 2018]. In fact, among all optimization-based rules, only PAV is proportional [Lackner and Skowron, 2018, Aziz et al., 2017].

However, when the unit cost assumption does not hold, PAV ceases to guarantee proportional representation. To see this, consider the city of Onetown shown in Figure 2. Onetown has 90,000 residents split in two districts, and has $90,000 available for participatory budgeting. The 60,000 residents of Leftside all vote for projects $\{L_1, L_2, L_3\}$ each of which costs $20,000. The 30,000 residents of Rightside vote for the single project $\{R\}$ which costs $45,000. Note that the prices are such that we can either afford to implement all three $L$-projects giving PAV score 110,000, or implement two $L$-projects and the $R$-project giving PAV score 120,000. Thus, PAV implements project $R$ and only two $L$-projects. However, note that Leftside residents form two thirds of the population of Onetown, and so by proportionality are entitled to two thirds of the budget ($60,000), which is enough to implement all three $L$-projects. Hence, Leftside is underrepresented by PAV.

To see what is going on, consider Twotown from Figure 2. Twotown is just like Onetown, except that now each projects cost $30,000. Note that for Twotown it is still the case that we can either afford all three $L$-projects, or two $L$-projects plus $R$. By the same calculation as before, PAV implements the latter possibility. This time, this is the proportional choice: Leftside now deserves only two projects, since only two projects are affordable with Leftside's share of the budget.

Onetown and Twotown are nearly identical: same number of residents, same district structure, same alternatives, same approval sets, and even the feasibility constraint (three $L$ or two $L$ plus $R$) is the same. Since the definition of PAV only depends on these characteristics, it must select the same outcome for both towns. But the prices differ, and therefore different outcomes are proportional, and hence PAV fails proportionality. The same is true for all other rules that depend only on preferences and feasibility constraints but not prices. This suggests that there is no variant of PAV that retains its proportionality guarantees beyond the unit cost case.

**Theorem 4.** *Every voting rule that only depends on voters' utility functions and the collection of budget-feasible sets must fail proportionality, even on instances with a district structure.* □

(An instance has *district structure* if projects and voters can be partitioned into disjoint districts such that each voter approves exactly the projects belonging to the voter's district.)

## B  Priceability of the Method of Equal Shares

Peters and Skowron [2020] introduced a concept called priceability, which imposes a certain kind of balance on a voting rule. Every rule that, like the Method of Equal Shares, equally splits the budget between voters and then sequentially purchases projects using the money of its supporters will be priceable. Priceability does not place any restrictions on how the rule splits the project's cost among supporters. The concept also allows initial budgets higher than 1; an outcome is priceable if there exists *some* budget limit for which it is priceable.

**Definition 12** (Priceability). A price system is a pair $\mathsf{ps} = (b, \{p_i\}_{i \in N})$, where $b \geq 1$ is the initial budget (where each voter controls equal part of the budget, namely $b/n$), and for each voter $i \in N$, $p_i \colon C \to \mathbb{R}_{\geq 0}$ is a *payment function* that specifies the amount of money a particular voter pays for the elected candidates.[9,10] An outcome $W$ is supported by a price system $\mathsf{ps} = (b, \{p_i\}_{i \in N})$ if the following conditions hold:

**(C1).** $u_i(c) = 0 \implies p_i(c) = 0$ for each $i \in N$ and $c \in C$,[11]

**(C2).** Each voter has the same initial budget of $b/n$ dollars: $\sum_{c \in C} p_i(c) \leq b/n$ for each $i \in N$.

**(C3).** Each elected candidate is fully paid: $\sum_{i \in N} p_i(c) = \mathrm{cost}(c)$ for each $c \in W$.

**(C4).** The voters do not pay for non-elected candidates: $\sum_{i \in N} p_i(c) = 0$ for each $c \notin W$.

**(C5).** For each unelected candidate $c \notin W$, the unspent budget of her supporters is at most $\mathrm{cost}(c)$: $\sum_{i \in N : u_i(c) > 0} \left( b/n - \sum_{c' \in W} p_i(c') \right) \leq \mathrm{cost}(c)$ for each $c \notin W$.

An outcome $W$ is said to be *priceable* if there exists a price system $\mathsf{ps} = (b, \{p_i\}_{i \in N})$ that supports $W$ (that satisfies conditions (C1)–(C5)).

It is known that the Method of Equal Shares is priceable in the approval-based committee-election model and in the general PB model this property is also preserved—indeed, the rule implicitly constructs the price system satisfying the above conditions.

## C  Exhaustiveness

A basic and very desirable efficiency notion is *exhaustiveness*, which requires that a voting rule spends its entire budget. Of course, due to the discrete model, we cannot guarantee that the rule will spend exactly 1 dollar (i.e., the entire budget); however, we can require that no additional project is affordable. Let us recall Definition 9:

**Definition** (Exhaustiveness, Aziz et al., 2018b). An election rule $\mathcal{R}$ is *exhaustive* if for each election instance $E$ and each non-selected candidate $c \notin \mathcal{R}(E)$ it holds that $\mathrm{cost}(\mathcal{R}(E) \cup \{c\}) > 1$.

Notably, MES fails to be exhaustive. It can happen that at the end of MES's execution, some project remains affordable, but the project's supporters do not have enough money to pay for it—the Method of Equal Shares then refuses to fund the project. For example, if we have two voters and two candidates, such that $v_1$ approves $\{c_1\}$ and $v_2$ approves $\{c_2\}$. Suppose that both candidates cost 1 dollar. Then MES returns $W = \emptyset$. In fact, it turns out that exhaustiveness is incompatible with priceability.

**Example 3.** We have 3 candidates and 3 voters. The first 2 voters approve $\{c_1\}$, and the third one approves $\{c_2, c_3\}$. We have $\mathrm{cost}(c_1) = 1$ and $\mathrm{cost}(c_2) = \mathrm{cost}(c_3) = 1/3$. The only exhaustive outcomes are $\{c_1\}$ and $\{c_2, c_3\}$. However, neither of them is priceable—indeed, to buy both $c_2$ and

---

[9]Peters and Skowron [2020] assumed that each voter is initially given one dollar, which corresponds to setting $b = n$, but that there is an additional variable that specifies the total price that needs to be paid for an elected candidate. These two formulations are equivalent, but the present definition seems more natural for PB.

[10]The requirement that $b \geq 1$ ensures that the voters have at least enough money to buy candidates with total cost 1 (that is, the value of the real budget). Without this requirement, an empty outcome $W = \emptyset$ would be priceable (with $b = 0$).

[11]While condition (C1) is well-justified in the approval-based setting, in the general PB model it is very weak. Indeed, the condition does not put any restrictions on the payments when $u_i(c)$ is very small, yet positive.

$c_3$, the third voter needs to control at least $^2/_3$ dollars. Then the first two voters control $^4/_3$ dollars and can buy candidate $c_1$, a contradiction. On the other hand, to buy $c_1$, the first two voters need to control at least 1 dollar. Then, the third voter controls at least $^1/_2$ dollars and buys $c_2$ or $c_3$, a contradiction. □

In some contexts, it may actually be a desirable feature of the Method of Equal Shares that it is not exhaustive, especially if unspent budget can be used in other productive ways (such as in next year's PB election). Arguably, in non-exhaustive examples, no remaining project has sufficient support to justify its expense; on that view, no further projects should be funded. In other situations, unspent budget may not be reusable, such as when the budget comes from a grant where unspent money needs to be returned (and the relevant decision makers do not obtain value from the grant-maker's alternative activities), or when the 'budget' is time (for example, when we use PB to plan activities for a day-long company retreat). In such situation, one might prefer an exhaustive rule.

Peters and Skowron [2020] proposed to complete the outcome elected by MES with the use of Phragmén's sequential rule (with initial budgets of the voters equal to the remainder left after running MES). We defined Phragmén's rule in the introduction. However, there is no obvious way of generalizing Phragmén's rule to non-approval utilities.[12]

However, in our opinion, it should preserve the main properties of this rule: priceability (as in Definition 12), committee monotonicity[13] and exhaustiveness. Unfortunately, it turns down that in the PB model:

1. priceability and exhaustiveness are mutually exclusive (Example 3),
2. any rule satisfying both outcome monotonicity and either exhaustiveness or priceability is strongly inefficient (Example 4).

**Example 4.** We have 2 voters and 2 candidates. Costs of the candidates are the following: $\text{cost}(c_1) = 1$, $\text{cost}(c_2) = 2$. Utilities of both voters gained from $c_1$ are equal to $\varepsilon$, and from $c_2$ are equal to 1. If the value of the voters' common budget is 1, then the only feasible outcomes are $\emptyset$ and $\{c_1\}$. If the rule satisfies either exhaustiveness or priceability, then it should elect $\{c_1\}$ in such case. However, if we increase voters' budget to 2, then outcome monotonicity requires that $c_1$ is still elected—and consequntly, $\{c_1\}$ is still the only possibility. However, in such case, $\{c_2\}$ is a clearly a much more efficient choice.

Since we have generalized the Method of Equal Shares to work for general additive valuations, there is another way for us to make it exhaustive. Recall that MES fails to be exhaustive in situations where the remaining projects' supporters do not have sufficient funds left. However, in elections where $u_i(c) > 0$ for all $i \in N$ and $C \in C$, every voter supports every candidate, and thus this problem never occurs. In fact, MES is exhaustive when run on profiles of this type.

**Proposition 3.** *Consider an election $E = (N, C, \text{cost}, \{u_i\}_{i \in N})$ such that $u_i(c) > 0$ for each $i \in N$ and $c \in C$. The outcome returned by the Method of Equal Shares for $E$ is exhaustive.*

*Proof.* For the sake of contradiction assume that an outcome $W$ returned by the Method of Equal Shares for an election instance $E$ is not exhaustive. Then, there exists a candidate $c \notin W$ such that $\text{cost}(W \cup \{c\}) \leq 1$. The voters paid in total $\text{cost}(W)$ dollars for $W$; their initial budget was 1, thus after $W$ is selected they all have at least $\text{cost}(c)$ unspent money. However, this means that at the end of the execution of MES there exists a possibly very large value of $\rho$ such that:

$$\sum_{i \in N} \min\left(\tfrac{1}{n} - p_i(W), u_i(c)\rho\right) = \sum_{i \in N} \left(\tfrac{1}{n} - p_i(W)\right) \geq \text{cost}(c).$$

Consequently, $c$ (or some other candidate) would be selected by MES, a contradiction. □

---

[12]One possibility, similar to our generalization of MES, would be to require that voters' payment for selected projects must be proportional to their utilities. Interestingly, this idea seems to not work at all for Phragmén's rule. For example, consider a committee election with two projects, $c_1$ and $c_2$, committee size $k = 1$, and two voters: The first voter assigns utility 1 to $c_1$ and utility 100 to $c_2$; the second voter assigns utility 1 to $c_1$ and 99 to $c_2$. When forced to use proportional payments, Phragmén's rule would choose $c_1$, a very inefficient choice.

[13]Intuitively, committee monotonicity says that if we increase the size of the budget then the new outcome must contain all the candidates that were selected given the old budget. This axiom can be naturally generalized to the PB model—there we will use term 'outcome montonicity'.

Thus, we can make MES exhaustive by perturbing the input utilities so that all utility values are positive; we call this strategy EXH1. Specifically, for a small $\epsilon > 0$ ($\epsilon \ll \min_{u_i(c)>0} u_i(c)$), and for each $i \in N, c \in C$ such that $u_i(c) = 0$ in the initial instance we set $u_i^\epsilon(c) = \epsilon$. Next, we run MES on the modified instance $\{u_i^\epsilon\}_{i \in N}$; by Proposition 3 the outcome is exhaustive. Finally we return the outcome identified by the Method of Equal Shares as $\epsilon \to 0$; the result is well-defined by the following result.

**Proposition 4.** *Consider any election $E = (N, C, \text{cost}, \{u_i\}_{i \in N})$. There exists some $\bar{\epsilon} > 0$ such that for all $0 < \epsilon_1, \epsilon_2 < \bar{\epsilon}$, the Method of Equal Shares returns the same outcome when run on $\{u_i^{\epsilon_1}\}_{i \in N}$ and $\{u_i^{\epsilon_2}\}_{i \in N}$.*

This process gives rise to a different voting rule, an exhaustive variant of the Method of Equal Shares. Note that this rule is not priceable since it may ask voters to pay for candidates that they assign utility 0. By necessity, this rule sometimes elects candidates that cannot be afforded by its supporters. In these cases, when we elect such a candidate $c$, we will ask all supporters of $c$ to pay all their remaining money for $c$, and split the remaining cost to be paid equally among voters who do not support $c$. Say that the maximum amount paid by a non-supporter for $c$ is $x$; the Method of Equal Shares selects those candidate that minimize the value $x$ at each step. Because voters are asked to spend their entire remaining budget if a non-affordable candidate they like is elected, this extension of the Method of Equal Shares will not distort the outcome too much.

Peters et al. [2021] suggested yet another method of completing outcomes returned by the Method of Equal Shares. This method, which we call EXHAUS2, was proposed in the context of approval ballots, but it naturally extends to additive utilities. In EXHAUS2 we run the Method of Equal Shares with the initial value of the budget $b_i$ set to a value possibly greater than the actual available budget, i.e., $b_i \geq 1$. Using a binary search we find the highest value of $b_i$ such that the total cost of the projects selected by MES does not exceed the actual budget. This method does not guarantee that the budget is exhausted (for a detailed discussion, see [Peters et al., 2021]), but in most cases this is indeed the case.

In Appendix H.1 we experimentally compare EXH1 and EXH2 based on data collected from real participatory budgeting instances. Our experiments show that EXH2 typically returns outcomes with higher values of total voters' satisfaction, and we suggest it as the preferred method.

# D   Omitted Proofs

## D.1   Proof of Theorem 2

**Theorem.** *Given an election $E$, by $u_{\max}$ we denote the highest utility a voter can get from a feasible outcome. Analogously, by $u_{\min}$ we denote the smallest, yet still positive utility a voter can get from a feasible outcome:*

$$u_{\max} = \max_{i \in N} \max_{\text{cost}(W) \leq 1} u_i(W) \qquad and \qquad u_{\min} = \min_{i \in N} \min_{u_i(W)>0} u_i(W).$$

*the Method of Equal Shares satisfies the $\alpha$-core property for $\alpha = 4 \log(2 \cdot u_{\max}/u_{\min})$.*

*Proof.* Towards a contradiction, assume there exist an election instance $E$, a winning outcome $W \in \mathcal{R}(E)$, a subset of voters $S \subseteq N$, and a subset of candidates $T \subseteq C$ with $\sum_{c \in T} \text{cost}(c) \leq |S|/n$ such that for each $i \in S$ and $c \in T$ it holds that $u_i(W \cup \{c\}) < u_i(T)/\alpha$.

Now, consider a fixed subset $S' \subseteq S$, and let:

$$\Delta(S') = \sum_{i \in S'} \big( u_i(T) - u_i(W) \big).$$

Similarly, as in the proof of Theorem 1, assume for a while that the voters from $S'$ have unrestricted initial budgets, and let us analyze how the Method of Equal Shares would proceed in such a case. By the pigeonhole principle it follows that in each step of the rule there exists a not-elected candidate $c \in T \setminus W$ such that:

$$\frac{u_{S'}(c)}{\text{cost}(c)} \geq \frac{\Delta(S')}{\text{cost}(T)}.$$

Indeed, if for each $c \in T \setminus W$ we had $\frac{u_{S'}(c)}{\text{cost}(c)} < \frac{\Delta(S')}{\text{cost}(T)}$, then:

$$\Delta(S') \leq \sum_{c \in T \setminus W} u_{S'}(c) < \sum_{c \in T \setminus W} \text{cost}(c) \cdot \frac{\Delta(S')}{\text{cost}(T)} \leq \Delta(S'),$$

a contradiction.

Thus, the price-per-utility that the voters pay for the selected candidates equals at most $\frac{\text{cost}(T)}{\Delta(S')}$. (This follows from the fact that the Method of Equal Shares selects candidates in such an order that the maximal price-per-utility the voters pay in a given round is minimized. The precise arguments are the same as in the proof of Theorem 1.) Now, consider the first moment when some voter in $S'$—call it $i$—uses more than its initial budget $1/n$. Until this time moment, the Method of Equal Shares behaves exactly in the same way as if the voters from $S'$ had their initial budgets set to $1/n$. Further, we know that in this moment, if we chose a candidate $c \in T$ that would be chosen if the voters had unrestricted budgets, then the voter $i$ would pay more than $1/n$ in total, and thus, would get the utility of more than $\frac{1}{n} \cdot \frac{\Delta(S')}{\text{cost}(T)}$. Since we assumed $u_i(W \cup \{c\}) < u_i(T)/\alpha$, we get that:

$$\frac{u_i(T)}{\alpha} > u_i(W) + u_i(c) > \frac{1}{n} \cdot \frac{\Delta(S')}{\text{cost}(T)}.$$

Since $\alpha \geq 2$, and so $u_i(T) - u_i(W) \geq u_i(T)/2$, we get that:

$$u_i(T) - u_i(W) \geq \frac{u_i(T)}{2} > \frac{\alpha \Delta(S')}{2n \cdot \text{cost}(T)}.$$

Let $S'' = S' \setminus \{i\}$. Clearly, we have that:

$$\Delta(S'') = \Delta(S') - (u_i(T) - u_i(W)) \leq \Delta(S') \left(1 - \frac{\alpha}{2n \cdot \text{cost}(T)}\right).$$

The above reasoning holds for each $S' \subseteq S$. Thus, we start with $S' = S$ and apply it recursively, in each iteration decreasing the size of $S'$ by 1. After $|S|/2$ iterations we are left with a subset $S_e$ such that:

$$\Delta(S_e) \leq \Delta(S) \left(1 - \frac{\alpha}{2n \cdot \text{cost}(T)}\right)^{\frac{|S|}{2}} \leq \Delta(S) \left(1 - \frac{\alpha}{2n \cdot \text{cost}(T)}\right)^{\frac{\text{cost}(T)n}{2}} < \Delta(S) \left(\frac{1}{e}\right)^{\frac{\alpha}{4}}.$$

Now, observe that $\Delta(S_e) \geq |S|/2 \cdot u_{\min}$ (for each $i \in S$ it must hold that $u_i(T) - u_i(W) > 0$) and that $\Delta(S) \leq |S| \cdot u_{\max}$. Thus, we get that:

$$\frac{|S|}{2} u_{\min} \cdot e^{\frac{\alpha}{4}} < |S| \cdot u_{\max},$$

which is equivalent to $e^{\frac{\alpha}{4}} < 2 \cdot \frac{u_{\max}}{u_{\min}}$ and, further, to $\alpha < 4 \log(2 \cdot u_{\max}/u_{\min})$. This gives a contradiction and completes the proof. □

## D.2 Proof of Proposition 2

Suppose that rule $\mathcal{R}$ satisfies FJR and take an $(\alpha, T)$-cohesive group of voters $S$ for some $\alpha \colon T \to [0; 1]$, $T \subseteq C$. For every voter $i \in S$ and every candidate $c \in T$ we have $u_i(c) \geq \alpha(c)$. We set $\beta = \sum_{c \in T} \alpha(c)$; clearly, we have also $u_i(T) \geq \beta$, thus $S$ is weakly cohesive. As $\mathcal{R}$ satisfies FJR, we have that $u_i(\mathcal{R}(E)) \geq \beta = \sum_{c \in T} \alpha(c)$, which completes the proof.

## D.3 Proof of Theorem 3

Suppose that there exists a weakly $(\beta, T)$-cohesive group $S$ which witnesses that FJR is not satisfied. Consider the voter $i \in S$ who was removed first by GCR and the outcome $W$ right after that step (since $S$ is weakly cohesive, such $i$ always exists). Since $i \in S$ and $S$ witnesses the FJR failure, we have $u_i(W) < \beta$. We know that $i$ was removed as a member of some weakly $(\beta', T')$-cohesive group $S'$. Just before $S'$ was removed, none of the members of $S$ had been removed. Thus, we have $\beta' \geq \beta$, as GCR maximizes this value. However, since $T' \subseteq W$, we have $\beta' \leq u_i(T') \leq u_i(W) < \beta$—a contradiction to $\beta' \geq \beta$. Hence, such a group $S$ does not exist.

# E Known Rules do not satisfy FJR

We show that the two major rules known to satisfy EJR for approval-based committee elections (the Method of Equal Shares and PAV) both fail FJR.

**Example 5** (Method of Equal Shares). Consider the following instance of approval-based committee elections for $n = 22$ voters, $m = 13$ candidates, and where the goal is to select a committee of size $k = 11$:

| | | | |
|---|---|---|---|
| voters 1-3: | $\{c_1, c_2, c_3, c_4, c_8\}$ | voters 13-15: | $\{c_1, c_2, c_3, c_4, c_{12}\}$ |
| voters 4-6: | $\{c_1, c_2, c_3, c_4, c_9\}$ | voters 16-18: | $\{c_5, c_6, c_7, c_8, c_9, c_{10}, c_{11}, c_{12}\}$ |
| voters 7-9: | $\{c_1, c_2, c_3, c_4, c_{10}\}$ | voters 19-21: | $\{c_5, c_6, c_7\}$ |
| voters 10-12: | $\{c_1, c_2, c_3, c_4, c_{11}\}$ | voter 22: | $\{c_{13}\}$. |

In the first 4 steps, MES chooses candidates $c_1, c_2, c_3, c_4$ (this happens for $\rho = 1/11 \cdot 15$). After that, each voter of the first 15 ones has $1/22 - 4/11 \cdot 15$ dollars. In next 3 steps, for $\rho = 1/11 \cdot 6$, candidates $c_5, c_6, c_7$ are chosen: 6 voters who support them spend all their money ($1/22 - 3/11 \cdot 6 = 0$). After that, the algorithm stops. Each of the first 15 voters has 4 candidates she approves; voters 16-18 approve 3 selected candidates. Thus, no member of the weakly $(5, \{c_1, c_2, c_3, c_4, c_8, c_9, c_{10}, c_{11}, c_{12}\})$-cohesive group of the first 18 voters has 5 representatives. □

**Example 6** (PAV). This example was first considered by Peters and Skowron [2020, Section 1]. We have $m = 15$ candidates and $n = 6$ voters, with the following preferences:

| | | | |
|---|---|---|---|
| voter 1: | $\{c_1, c_2, c_3, c_4\}$ | voter 4: | $\{c_7, c_8, c_9\}$ |
| voter 2: | $\{c_1, c_2, c_3, c_5\}$ | voter 5: | $\{c_{10}, c_{11}, c_{12}\}$ |
| voter 3: | $\{c_1, c_2, c_3, c_6\}$ | voter 6: | $\{c_{13}, c_{14}, c_{15}\}$. |

The size of the committee to be elected is $k = 12$. PAV chooses in this case committee $\{c_1, c_2, c_3, c_7, c_8, c_9, c_{10}, c_{11}, c_{12}, c_{13}, c_{14}, c_{15}\}$. Hence, no voter from the weakly $(4, \{c_1, c_2, c_3, c_4, c_5, c_6\})$-cohesive group consisting of the first 3 voters has 4 representatives. □

# F Additional Results about the Greedy Cohesive Rule

## F.1 Priceability and Exhaustiveness of GCR

GCR satisfies neither priceability (Definition 12) nor exhaustiveness (Definition 9). However, we will prove that an outcome elected by this rule can be always completed to a priceable one; this suggests that GCR never elects outcomes that are "too unbalanced". In the proof of Theorem 5 we describe precisely how such a completion can be implemented. Using a somewhat different completion scheme, we can complete GCR to an exhaustive outcome (using the exhaustive variant of MES). This way we obtain an outcome that is both exhaustive and also close to being priceable.

We start by proving three useful lemmas.

**Lemma 1** (Polyandrious generalization of Hall's marriage theorem). *Let $G = (U + V, E)$ be a bipartite graph and for every $A \subseteq U$ denote by $N_G(A)$ the neighbourhood of $A$, i.e. $N_G(A) = \{v \in V : \exists u \in A.\{u, v\} \in E\}$. Let $q \in \mathbb{N}$. Then for each $A \subseteq U$ we have that $|N_G(A)| \geq |A| \cdot q$ if and only if there exists a one-to-$q$ mapping from each vertex in $U$ to some $q$ vertices in $V$, such that to each vertex from $V$ at most one vertex from $U$ is mapped.*

*Proof.* Consider the graph $G'$ obtained by replacing set $U$ with its $q$ copies: $U_1, \ldots, U_q$ (we also copy edges). Consider now any set $A \in \bigcup_i U_i$. As $\bigcup_i U_i$ consists of $q$ separate copies, there exists $i \in [q]$ such that $|A \cap U_i| \geq |A|/q$. Hence, from our assumption we have that $|N_G(A)| \geq |N_G(A \cap U_i)| \geq q|A \cap U_i| \geq |A|$. Now, from Hall's marriage theorem, in $G'$ there exists a matching between $\bigcup_i U_i$ and $V$, covering set $\bigcup_i U_i$. Hence, in graph $G$ it is enough to map each vertex $u \in U$ to these $q$ vertices in $V$, to which $q$ copies of $u$ are matched in $G'$. Naturally, the implication holds also in the reverse direction—if there exists a one-to-$q$ mapping in $G$ as described above, then trivially for all $A \subseteq U$ we have that $|N_G(A)| \geq q|A|$. □

**Lemma 2.** *Let $S$ be $n$ $(\beta, T)$-cohesive group which is selected in some step of GCR. For every subset $A \subseteq T$, the size of the set of voters $S' := \{i \in S : u_i(A) > 0\}$ is at least $\mathrm{cost}(A) \cdot n$.*

*Proof.* The statement is trivial for $\mathrm{cost}(A) = 0$, so assume that $\mathrm{cost}(A) > 0$. Suppose for the sake of contradiction that the set $S' \subseteq S$ defined above has smaller size than $\mathrm{cost}(A) \cdot n$. Then group $S \setminus S'$ together with set $T \setminus A$ is $(\beta, T \setminus A)$-cohesive. Indeed,

$$|S \setminus S'| \geq |S| - \mathrm{cost}(A) \cdot n \geq \mathrm{cost}(T) \cdot n - \mathrm{cost}(A) \cdot n = \mathrm{cost}(T \setminus A) \cdot n.$$

Thus, as $\mathrm{cost}(A) > 0$, we have $\mathrm{cost}(T \setminus A) < \mathrm{cost}(T)$. Thus, GCR would select $S \setminus S'$ instead of group $S$, a contradiction. $\qquad\square$

**Lemma 3.** *For every outcome of the GCR rule, there always exists a payment function satisfying conditions (C1)–(C4) with $b = 1$.*

*Proof.* Consider a single step of GCR and let $S$ be an $(\beta, T)$-cohesive group considered in that step. We will prove that there exists a price system in which voters from $S$ pay $\mathrm{cost}(c)$ dollars for each candidate $c \in T$[14].

Denote by $d$ the least common multiple of the denominators of the rational numbers from the set: $\{\mathrm{cost}(c) : c \in T\}$. Note that $1/d$ is a divisor of all these costs. Assume that each candidate $c$ is splitted into $\mathrm{cost}(c) \cdot d$ parts, each one associated with cost $1/d$. Besides, assume that each voter has $d$ *coins*, each one worth $1/d \cdot n$ dollars.

Consider the bipartite graph $G = (A_T + A_S, E)$, where $A_S$ is the set of all voters' coins and $A_T$ is the set of all candidates' parts. In $G$ there is an edge between a coin of a voter $i \in S$ and a part of a candidate $c \in T$ if and only if $u_i(c) > 0$.

Now, consider a set $A \subseteq A_T$, and let us assess the size of the neighbourhood $N_G(A)$. Let $C(A)$ denote the set of candidates whose some parts belongs to $A$. There is an edge from $p \in A$ to a coin of a voter $i$ only if $i$ assigns a positive utility to the candidate of $p$. Thus, $N_G(A)$ consists of coins of those voters, who assign a positive utility to some candidate from $C(A)$. By Lemma 2 there are at least $\mathrm{cost}(C(A)) \cdot n$ such voters, each voter comes with $d$ coins, thus:

$$|N_G(A)| \geq \mathrm{cost}(C(A)) \cdot n \cdot d \geq \mathrm{cost}(A) \cdot n \cdot d.$$

Further, since each part of $A$ costs exactly $1/d$, we get that:

$$|N_G(A)| \geq \mathrm{cost}(A) \cdot n \cdot d = 1/d \cdot |A| \cdot n \cdot d = |A| \cdot n.$$

Hence, from Lemma 1 we have that there exists a mapping from $A_T$ to $A_S$ such that every part of every candidate $c$ is mapped to $n$ coins and to each coin at most one candidate is mapped.

Now the payment function is constructed as follows: for every voter $i \in S$ and candidate $c \in T$, if exactly $q$ coins of $i$ are mapped with some parts of $c$, then $p_i(c) = q/d \cdot n$. It is straightforward to check that such a payment function satisfies conditions (C1)–(C4) for $b = 1$, which completes the proof. $\qquad\square$

Finally, we can state the main result of this subsection.

**Theorem 5.** *Every outcome $W$ elected by GCR can be completed to some priceable outcome.*

*Proof.* From Lemma 3, we know that there exists a family of payment functions $\{p\}_{i \in N}$ satisfying conditions (C1)–(C4) for $W$. Now, to obtain outcome $W'$ supported by a valid price system, it is enough to run MES for this instance with initial outcome set to $W$ and the initial endowment of every voter $i \in N$ set to $1/n - \sum_{c \in W} p_i(c)$. $\qquad\square$

### F.2 Some Drawbacks of the Greedy Cohesive Rule

Since GCR satisfies FJR but MES does not, we may conclude that GCR is a better rule. Clearly, GCR is custom-engineered to satisfy FJR. Thus, we may expect the rule to be deficient in other dimensions. The results presented in Appendix F.1 certainly suggest that GCR is not pathological, but in this section we consider some examples where MES seems to select better outcomes than GCR.

---

[14]Note that if a candidate $c \in T$ has been considered in previous steps of the algorithm, she does not need to be paid again. However, this case would even strengthen the proof (we could just not charge voters assigned to paying for her and conditions (C1)–(C4) would be satisfied), so further we assume that $T$ contains only candidates not elected yet.

We begin by discussing a property that Peters and Skowron [2020] call *laminar proportionality*. This property identifies a family of well-behaved preference profiles and specifies the outcome on those profiles. The axiom is defined for the case of approval-based committee elections. MES satisfies; the following example shows that GCR does not.

**Example 7** (GCR fails laminar proportionality). Let $N = \{1, 2, 3, 4\}$ and $k = 8$, and introduce the candidate sets $X = \{c_1, \ldots, c_4\}$, $Y = \{c_5, \ldots, c_{10}\}$, and $Z = \{c_{11}, c_{12}\}$. The first three voters approve $X \cup Y$, and the fourth one approves $X \cup Z$. Two copies of the profile are depicted below. The candidates are represented by boxes; each candidate is approved by the voters who are below the corresponding box.

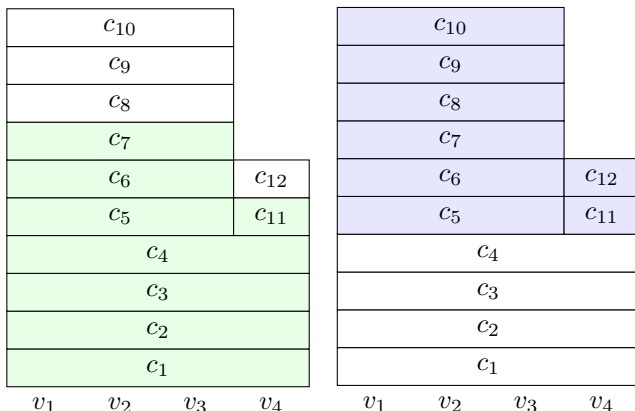

In this election instance, laminar proportionality would require that the voting rule selects all the candidates from $X$ since they are approved by everyone. After electing the candidates in $X$, four seats are left to fill. Since the group $\{v_1, v_2, v_3\}$ the three times as large as the group $\{v_4\}$, laminar proportionality requires that we elect three candidates from $Y$ and one candidate from $Z$. Thus, the committee indicated by the green boxes on the left-hand figure is laminar proportional.

On the other hand, in the first step GCR can choose the weakly $(6, Y)$-cohesive group $\{v_1, v_2, v_3\}$ and in the second step it can select the weakly $(2, Z)$-cohesive group $\{v_4\}$. This results in the blue committee depicted in the right-hand figure; this committee fails laminar proportionality. □

Example 7 shows that in general, GCR is not laminar proportional, as it can return committees which are prohibited by the axiom. However, this example is not fully satisfactory, as it depends on tie-breaking. For example, in the first step we could choose the weakly $(6, \{c_2, c_3, c_4, c_5, c_6\})$-cohesive group containing the first three voters, and in the second step the weakly $(2, \{c_1, c_{11}\})$-cohesive group containing the last voter. An open question is whether GCR can always elect a committee satisfying laminar proportionality (among others). However, the following example shows that for some 'nearly laminar' instances, GCR does not match the general intuition standing behind this axiom.

**Example 8.** Modify the instance described in Example 7 in the following way: we have $N = [4000]$. Voter 1 approves only candidates from $Y$, voters 2 to 3000 approve $X \cup Y$, voters 3001 to 3999 approve $X \cup Z$ and voter 4000 approves $Z$.

This instance is not laminar (because of the two voters not approving $X$), but it is close to being laminar and it is reasonable to expect that the elected committee should be the same as the green one from Example 7. The Method of Equal Shares uniquely elects that committee. On the other hand, GCR selects first the weakly $(6, Y)$-cohesive group containing the first 3000 voters and in the second step the weakly $(2, Z)$-cohesive group containing the last 1000 voters. After that the algorithm stops, electing committee $Y \cup Z$, as depicted above. Note that in this case, the choice of weakly cohesive groups is unique.   □

Examples 7 and 8 do not rule out the existence of an FJR rule that is also laminar proportional; the existence of a natural example of such a rule is an interesting open problem.

## G   Method of Equal Shares and GCR for Ordinal Ballots

In this section we discuss how our two rules can be adapted for committee elections where voters have *ordinal preferences*, that is, voters express their preferences by ranking the candidates. The main idea is straightforward: we convert voters' preference rankings into additive valuations, by using positional scoring rules, and then apply our rules to the resulting election. Note that if we use scoring rules that assign positive values to positions in voters' rankings, we always obtain exhaustive rules. We will show that when we use a lexicographic conversion scheme in which voters care infinitely more about their top-ranked candidate than their second-ranked candidate and so on, then the Method of Equal Shares satisfies an axiom called Proportionality for Solid Coalitions (PSC). which was first introduced to analyze the Single Transferable Vote [Woodall, 1994, Tideman and Richardson, 2000]. (GCR does not satisfy PSC.) The Method of Equal Shares as applied to ordinal preferences is related to the Expanding Approvals rule of Aziz and Lee [2020]. Interestingly, due to its flexibility, MES can be used to extend the proportionality idea behind PSC beyond a lexicographic interpretation of preferences: Depending on how we convert voters' preference rankings to utilities, we obtain different forms of proportionality (cf., Faliszewski et al., 2019). For example, if we use Borda scores, the rule chooses outcomes where the average position of selected candidates in voters' rankings is high.

### G.1   Model for Ordinal Preferences

In this section we assume that each voter $i \in N$ submits a strict preference order $\succ_i$ over the set of candidates. The order $c_{i_1} \succ_i c_{i_2} \succ \ldots \succ c_{i_m}$ means that $c_{i_1}$ is voter's $i$ most preferred candidate, $c_{i_2}$ is her second most-preferred candidate, and so on. By $\mathrm{pos}_i(c)$ we denote the position of candidate $c$ in $i$'s preference ranking. In the above example we have $\mathrm{pos}_i(c_{i_1}) = 1$, $\mathrm{pos}_i(c_{i_2}) = 2$, and so on. For sets $A$ and $B$, we write $A \succ_i B$ if $a \succ_i b$ for all $a \in A, b' \in B$.

Further, we assume unit costs, so that the goal is to select a committee of $k$ candidates, and thus that the cost of each candidate is $1/k$.

**Definition 13** (Proportionality for Solid Coalitions (PSC)). An outcome $W$ satisfies PSC if for each $\ell \in [k]$, each subset of voters $S \subseteq N$ with $|S| \geq n\ell/k$, and each subset of candidates $T$ such that $T \succ_i C \setminus T$ for all $i \in S$, it holds that $|W \cap T| \geq \min(\ell, |T|)$.

A rule satisfies PSC if for each election it only returns outcomes that satisfy PSC.

Definition 13 focuses on voters' top preferences—intuitively, it requires that if $c \succ_i c'$, then the utility that voter $i$ assigns to candidate $c$ is infinitely higher than that assigned to $c'$. MES naturally extends to such preferences, which we call *lexicographic utilities*, but we need to adapt Definition 1 to use a slightly different interpretation of the price per unit of utility, $\rho$. So far we assumed that $\rho$ is a positive real value; in order to adapt the definition to lexicographic preferences we assume that $\rho \in [m]$, and that the multiplication by candidates' utilities is defined as follows:

$$\rho \cdot u_i(c) = \begin{cases} 1 & \text{if } \rho \geq \text{pos}_i(c), \\ 0 & \text{otherwise.} \end{cases}$$

**Proposition 5.** *The Method of Equal Shares for lexicographic utilities satisfies PSC.*

*Proof.* Consider a committee $W$ returned by the Method of Equal Shares for an election instance $(N, C, k, \{\succ_i\}_{i \in N})$. Let $\ell \in [k]$, $S \subseteq N$, and $T$ be as in Definition 13. The voters in $S$ initially have the following budget:

$$|S| \cdot 1/n \geq n\ell/k \cdot 1/n = \ell/k.$$

Consider the steps of MES as the price per unit of utility, $\rho$, increases from 1 to $|T|$. In each such step, each voter from $S$ can pay only for the candidates in $T$. Indeed, each candidate $c \in C \setminus T$ occupies a worse position than $|T|$ in those voters' preference rankings, and so for each $i \in S$ we have $\rho \cdot u_i(c) = 0$ (since $\rho \leq |T|$). When $\rho$ reaches $|T|$, then for each candidate $c \in T$ and each $i \in S$ we have $u_i(c) = 1$. The voters from $S$ have enough money to buy $\ell$ candidates, and so they will buy at least $\min(\ell, |T|)$ candidates from $T$. $\square$

One may wonder, given the lexicographic utility scheme, whether PSC is just a consequence of the Method of Equal Shares satisfying EJR. Example 9 below shows that this is not the case and that the two axioms are logically incomparable in this context. FJR and PSC are also logically incomparable.

**Example 9.** Consider three voters with the following preference orders over $C = \{c_1, c_2, c_3, c_4\}$:

$$1\colon c_1 \succ c_2 \succ c_3 \succ c_4$$
$$2\colon c_2 \succ c_3 \succ c_1 \succ c_4$$
$$3\colon c_3 \succ c_1 \succ c_2 \succ c_4.$$

Assume $k = 2$. In this example PSC would require that two candidates from $\{c_1, c_2, c_3\}$ are elected. On the other hand, committee $\{c_1, c_4\}$ satisfies FJR.

Now, consider two voters with the following preferences:

$$1\colon c_1 \succ c_2 \succ c_3 \succ c_4$$
$$2\colon c_4 \succ c_1 \succ c_3 \succ c_2.$$

Assume $k = 1$. Here, EJR requires that $c_1$, $c_2$, or $c_4$ must be selected. On the other hand, $\{c_3\}$ is a committee that satisfies PSC. $\square$

GCR can also be adapted to lexicographic utilities. In this case, it is sufficient to assume that the utilities are exponentially decreasing with the positions—for each $i \in N$ and $c \in C$ we set $u_i(c) = m^{-\text{pos}_i(c)}$. Then, for each $c$ we have that $u_i(c) > \sum_{c' \prec_i c} u_i(c')$, and so the utility a voter assigns to a candidate in position $p$ is higher than the utility that it would assign to any committee all of whose members are ranked below $p$.

**Proposition 6.** *GCR for lexicographic utilities fails PSC.*

*Proof.* We show that GCR fails PSC. Consider the following preference profile:

$$1\colon c_1 \succ c_7 \succ c_8 \succ c_6 \succ c_4 \succ c_5 \succ c_2 \succ c_3 \succ c_9 \succ c_{10} \succ c_{11} \succ c_{12}$$
$$2\colon c_1 \succ c_7 \succ c_8 \succ c_6 \succ c_4 \succ c_5 \succ c_2 \succ c_3 \succ c_9 \succ c_{10} \succ c_{11} \succ c_{12}$$
$$3\colon c_1 \succ c_2 \succ c_3 \succ c_6 \succ c_4 \succ c_5 \succ c_7 \succ c_8 \succ c_9 \succ c_{10} \succ c_{11} \succ c_{12}$$
$$4\colon c_1 \succ c_2 \succ c_3 \succ c_6 \succ c_4 \succ c_5 \succ c_7 \succ c_8 \succ c_9 \succ c_{10} \succ c_{11} \succ c_{12}$$
$$5\colon c_1 \succ c_2 \succ c_3 \succ c_6 \succ c_4 \succ c_5 \succ c_7 \succ c_8 \succ c_9 \succ c_{10} \succ c_{11} \succ c_{12}$$
$$6\colon c_1 \succ c_2 \succ c_3 \succ c_6 \succ c_4 \succ c_5 \succ c_7 \succ c_8 \succ c_9 \succ c_{10} \succ c_{11} \succ c_{12}$$
$$7\colon c_2 \succ c_3 \succ c_1 \succ c_7 \succ c_8 \succ c_4 \succ c_5 \succ c_6 \succ c_9 \succ c_{10} \succ c_{11} \succ c_{12}$$
$$8\colon c_3 \succ c_2 \succ c_1 \succ c_7 \succ c_8 \succ c_4 \succ c_5 \succ c_6 \succ c_9 \succ c_{10} \succ c_{11} \succ c_{12}$$
$$9\colon c_4 \succ c_5 \succ c_9 \succ c_7 \succ c_8 \succ c_1 \succ c_2 \succ c_3 \succ c_6 \succ c_{10} \succ c_{11} \succ c_{12}$$
$$10\colon c_5 \succ c_4 \succ c_9 \succ c_7 \succ c_8 \succ c_1 \succ c_2 \succ c_3 \succ c_6 \succ c_{10} \succ c_{11} \succ c_{12}$$
$$11\colon c_{10} \succ c_{11} \succ c_{12} \succ c_7 \succ c_8 \succ c_1 \succ c_2 \succ c_3 \succ c_4 \succ c_5 \succ c_6 \succ c_9$$
$$12\colon c_{11} \succ c_{10} \succ c_{12} \succ c_7 \succ c_8 \succ c_1 \succ c_2 \succ c_3 \succ c_4 \succ c_5 \succ c_6 \succ c_9.$$

Assume $k = 4$. Here, GCR will first pick $S = \{1, \ldots, 6\}$ as a weakly cohesive group, with the corresponding set of candidates $T = \{c_1, c_6\}$. Indeed, if $T$ consisted of 3 candidates, then $S$ would need to have at least 9 voters. However, any 9 voters rank at least 4 different candidates at the top position, thus at least one of them would have a lower satisfaction than the voters from $S$ have from $T$. By the same argument, $T$ cannot consist of 4 candidates. If $T$ consisted of 2 candidates but $S$ included one voter from $7, \ldots, 12$, then the satisfaction of voter 1 or 2 would also be lower. Indeed, these two voters rank $c_2$, $c_3$, $c_4$, $c_5$, $c_{10}$, and $c_{11}$ (that is candidates that appear in the top positions) below $c_6$.

Hence, GCR picks $c_1$ and $c_6$, and removes the first 6 voters from further consideration. In the second step, the rule picks $c_7$ and $c_8$. This is because each other candidate appears at most twice before $c_7$ and $c_8$ in the remaining voters' rankings. Thus, the rule picks $c_1, c_6, c_7$ and $c_8$.

On the other hand, by looking at voters $3, \ldots, 8$ we observe that PSC requires that two candidates from $c_1, c_2, c_3$ should be selected. □ □

## H  Experiments

In this section we look at data describing voters' preferences, collected from participatory budgeting elections carried out in several major cities in Poland. The data is available publicly under the following URL: http://pabulib.org/ under CC-BY-SA license. It has been collected and described by [Stolicki et al., 2020]. Based on this data, we make a number of observations regarding election rules that we discuss in this paper.

In election instances that we have considered the projects were divided into several groups. One group consisted of city-wide projects, and each other group contained projects that were assigned to one of several city districts. Each voter was allowed to approve at most ten city-wide projects, and at most ten projects from her district. A part of the municipal budget was assigned to city-wide projects and the other part was divided among the districts, proportionally to their populations. Currently, the cities that we consider use a rule that selects projects greedily until the budget is exhausted; the projects are picked in the order of the number of garnered approval votes.

In our experimental analysis we used two types of voters' preferences:

**Approval utilities:** corresponding directly to the approval-ballots from our PB data.

**Cardinal utilities:** for each voter $i$ and each project $c_j$ we obtained the value of the utility $u_i(c_j)$ as follows. If $i$ does not approve $c_j$, we assume that $u_i(c_j) = 0$. If $i$ approves $c_j$, we sample $u_i(c_j)$ from the normal distribution centred at $\mathrm{cost}(c_j)$. We also tested similar assumptions, where the values $u_i(c_j)$ were sampled from the uniform and exponential distributions, but the results led to qualitatively consistent conclusions.

The rules that we are interested in comparing are the Method of Equal Shares, the greedy approval rule, which is a rule currently used for selecting projects, Phragmén's rule, and the sequential version

of PAV (sPAV)[15]. We have limited the experiments to polynomial-time algorithms, since the instances are too large to efficiently run e.g. the non-sequential versions of Phragmén's rule or PAV for them. Since Phragmén's rule does not extend to cardinal utilities, we assume that this rule is used only for approval utilities.

In our experiments we apply the greedy approval rule in exactly the same way as it is used by the cities, that is this rule is used separately in each district. In contrast, we run the Method of Equal Shares, Phragmén's rule, and sPAV on instances constructed by merging district-wide elections. This way, we analyse whether the considered rules would make proportional choices, even if the projects were not preassigned to specific districts.

In our analysis we used the following metrics:

**Total utility (UTIL).** The total utility of the voters from the selected set of projects $W$, that is $\sum_{i \in N} \sum_{c \in W} u_i(c)$.

**Distribution of projects (PROJ-DIS).** For each election instance we look at the projects selected from each district. We compute their cost and divide it by the fraction of the budget that is proportional to the population of the district (excluding city-wise projects that have been selected). From those ratios we take a variance.

**Distribution of utilities (UTIL-DIS).** For each election instance and each voter $i$ we compute her normalised utility from the set of selected projects $W$, which we define as $\sum_{i \in N} \sum_{c \in W} u_i(c)$ divided by $n \cdot \sum_{c \in W} u_i(c)$. We compute the variance of these values.

We normalise each of these three metrics: when computing the value of the metric for a committee $W$ we divide it by the value of the metric for the committee returned by the greedy approval rule.

First we checked whether the outcome that was in fact selected by the cities is fair according to the kind of fairness criteria we have been studying. We found that in at least 59 out of 366 elections (16%), EJR was failed. In most cases, the failure was of the form that there was a group of voters who approved 0 of the selected projects but who approved an unelected project in common, and the group was large enough to afford that project. Next, we checked the following two hypothesis.

### H.1 Comparing Exhaustive Variants of the Method of Equal Shares

We compare three different ways of making the Method of Equal Shares exhaustive. The first two methods, EXH1 and EXH2 were described in Appendix C. The third method, EXH3, uses the utilitarian strategy: we first select projects using the Method of Equal Shares, and then add projects greedily, in each round selecting a project that maximises the ratio of the total utility to the cost and that fits in the remaining budget.

The results of our experiments for approval utilities are presented in Table 1. We make the following conclusions:

1. Different exhaustive variants of the Method of Equal Shares vary significantly. MES gives a lot of flexibility to a mechanism designer, selecting often smaller outcomes, yet still satisfying strong fairness requirements, such as EJR. Depending on the specific objectives, a mechanism designer can decide to complete this outcome using different strategies. If the total utility is a primary criterion, then using variant EXH3 gives considerably better results than the other two variants.

2. Both EXH1 and EXH2 produce outcomes that balance voters' satisfaction much better than the current solution (in this sense, both of them are more proportional). However, the outcomes produced by EXH2 are qualitatively better and divide the budget between districts in a substantially fairer way than the ones produced by EXH1. Therefore we suggest EXH2 as the preferred method, when proportionality is the primary criterion.

we checked whether the outcome that was in fact selected by the cities is fair according to the kind of fairness criteria we have been studying. We found that in at least 59 out of 366 elections (16

---

[15]In fact, instead of PAV we use a rule that greedily maximizes smoothed Nash welfare, i.e., a rule that in every round picks a candidate $c$ maximizing $\sum_{i \in N} \log(1 + u_i(W \cup \{c\}))$, where $W$ is a committee elected so far. In such form, the rule has a straightforward generlization to additive utilities.

| Election | UTIL | | | UTIL-DIS | | | PROJ-DIS (VAR) | | |
|---|---|---|---|---|---|---|---|---|---|
| | EXH1 | EXH2 | EXH3 | EXH1 | EXH2 | EXH3 | EXH1 | EXH2 | EXH3 |
| | | | | Approval utilities | | | | | |
| Cracow-18 | 1.28 | 1.39 | 1.68 | 0.67 | 0.64 | 0.58 | 1.61 | 1.39 | 11.11 |
| Cracow-19 | 1.33 | 1.61 | 1.9 | 0.8 | 0.66 | 0.62 | 2.71 | 1.86 | 17.57 |
| Cracow-20 | 1.11 | 1.5 | 1.68 | 0.99 | 0.69 | 0.66 | 2.9 | 2.47 | 12.35 |
| Czestochowa-20 | 0.8 | 1.13 | 1.16 | 1.23 | 0.9 | 0.92 | 5.45 | 5.34 | 7.09 |
| Warsaw-17 | 1.26 | 1.36 | 1.54 | 1.11 | 0.91 | 0.99 | 41.77 | 7.37 | 26.09 |
| Warsaw-18 | 1.33 | 1.43 | 1.69 | 1.01 | 0.9 | 0.98 | 34.05 | 8.73 | 54.89 |
| Warsaw-19 | 1.33 | 1.37 | 1.63 | 1.24 | 1.07 | 1.25 | 35.98 | 6.98 | 50.48 |
| | | | | Cardinal utilities | | | | | |
| Cracow-18 | 0.25 | 0.39 | 1.69 | 0.42 | 0.39 | 0.66 | 1.64 | 1.5 | 58.13 |
| Cracow-19 | 0.29 | 0.46 | 1.8 | 0.46 | 0.33 | 0.62 | 2.8 | 1.9 | 4.58 |
| Cracow-20 | 0.21 | 0.61 | 2.0 | 0.29 | 0.28 | 0.38 | 2.78 | 2.98 | 4.16 |
| Czestochowa-20 | 0.08 | 1.33 | 1.5 | 0.74 | 0.76 | 0.67 | 4.68 | 6.46 | 66.12 |
| Warsaw-17 | 0.63 | 1.07 | 2.19 | 0.56 | 0.6 | 1.69 | 63.01 | 13.21 | 361.12 |
| Warsaw-18 | 0.8 | 1.16 | 2.24 | 0.72 | 0.7 | 1.67 | 41.28 | 18.18 | 432.02 |
| Warsaw-19 | 0.68 | 1.07 | 1.98 | 0.73 | 0.66 | 1.09 | 55.11 | 24.81 | 316.19 |

Table 1: The results of the experiments comparing exhaustive variants of the Method of Equal Shares.

## H.2 Comparing the Method of Equal Shares, Phragmén's rule, and PAV

In our second set of experiments we have compared MES with Phragmén's rule, and PAV. The results of those experiments are presented in Table 2. We make the following observations:

1. For approval utilities, the Method of Equal Shares and Phragmén's rule give very similar results. According to the utilitarian criterion, and to the distribution of utilities, these results are better than the results returned by the currently used method. Also, those rules divide the budget proportionally among the districts (the variance in the PROJ-DIS criterion is relatively low). PAV is considerably worse, both in terms of the total utility of the selected projects, and in terms of proportionality.

2. For cardinal utilities we observe a difference between MES and Phragmén's rule. This difference is expected since Phragmén's rule does not take into account the more fine-grained information on utilities, but operates only on approval ballots. On the other hand, our results suggest that there is indeed a considerable advantage of using rules that use cardinal utilities. We further observe that for cardinal utilities PAV returns outcomes with a higher total utility, but this happens at the huge cost of proportionality. The Method of Equal Shares always selects proportional outcomes, and outperforms Phragmén's rule with respect to the utilitarian criterion.

| Election | UTIL | | | UTIL-DIS (VAR) | | | PROJ-DIS (VAR) | | |
|---|---|---|---|---|---|---|---|---|---|
| | **RX** | **PHRAG** | **PAV** | **RX** | **PHRAG** | **PAV** | **RX** | **PHRAG** | **PAV** |
| Approval utilities | | | | | | | | | |
| Cracow-18 | 1.39 | 1.39 | 0.38 | 0.64 | 0.64 | 1.95 | 1.39 | 1.39 | 19.14 |
| Cracow-19 | 1.61 | 1.6 | 0.64 | 0.66 | 0.65 | 1.36 | 1.86 | 1.65 | 1.15 |
| Cracow-20 | 1.5 | 1.5 | 0.77 | 0.69 | 0.7 | 0.99 | 2.47 | 2.49 | 1.29 |
| Czestochowa-20 | 1.13 | 1.1 | 0.83 | 0.9 | 0.96 | 1.05 | 5.5 | 4.41 | 20.97 |
| Warsaw-17 | 1.36 | 1.37 | 0.4 | 0.91 | 0.93 | 1.19 | 7.37 | 6.24 | 86.22 |
| Warsaw-18 | 1.43 | 1.44 | 0.47 | 0.9 | 0.91 | 1.4 | 8.73 | 10.14 | 102.03 |
| Warsaw-19 | 1.4 | 1.41 | 0.37 | 1.07 | 1.1 | 1.32 | 6.98 | 8.59 | 83.46 |
| Warsaw-20 | 1.41 | 1.42 | 0.91 | 0.89 | 0.91 | 0.96 | 1.14 | 1.16 | 0.24 |
| Warsaw-21 | 1.62 | 1.64 | 0.93 | 0.73 | 0.8 | 0.84 | 1.06 | 1.09 | 0.19 |
| Cardinal utilities | | | | | | | | | |
| Cracow-18 | 0.39 | 0.28 | 1.46 | 0.39 | 0.37 | 0.72 | 1.49 | 1.28 | 33.98 |
| Cracow-19 | 0.46 | 0.4 | 1.55 | 0.33 | 0.36 | 0.81 | 1.94 | 1.78 | 1.11 |
| Cracow-20 | 0.61 | 0.59 | 1.83 | 0.28 | 0.29 | 0.44 | 3.08 | 2.33 | 1.16 |
| Czestochowa-20 | 1.33 | 0.47 | 1.63 | 0.75 | 1.15 | 0.66 | 6.33 | 4.28 | 57.07 |
| Warsaw-17 | 1.07 | 0.92 | 1.43 | 0.58 | 0.56 | 1.45 | 13.67 | 12.33 | 154.05 |
| Warsaw-18 | 1.17 | 0.96 | 1.4 | 0.7 | 0.69 | 1.36 | 18.86 | 14.35 | 118.95 |
| Warsaw-19 | 1.06 | 0.84 | 1.36 | 0.66 | 0.71 | 1.2 | 24.78 | 17.08 | 179.77 |
| Warsaw-20 | 1.01 | 0.79 | 1.28 | 0.83 | 0.91 | 0.89 | 1.16 | 1.13 | 15.09 |
| Warsaw-21 | 1.19 | 0.68 | 1.35 | 0.58 | 0.45 | 0.73 | 1.13 | 1.04 | 10.13 |

Table 2: The results of the experiments comparing the Method of Equal Shares, Phragmén's rule, and PAV.