# OpenReview forum: "Proportional Participatory Budgeting with Additive Utilities"
_NeurIPS.cc/2021/Conference — NeurIPS 2021 Poster_

### Official Review · Reviewer_T7rt · 2021-07-05

**Rating:** 8
**Confidence:** 4

**Summary:**

The authors introduce two axioms for participatory budgeting, EJR and FJR, which are related to proportionality/fairness of PB mechanisms. Their main result shows that a proposed PB mechanism, "Rule X" satisfies EJR, and PAV (an incumbent mechanism), and its variants, do not. They briefly describe a rule (Greedy Cohesive) that satisfies FJR. They describe some experiments with real PB data, which are not included in the main paper.


**Ethical Concerns:**

Nothing in addition to my previous comments.


**Limitations And Societal Impact:**

There is very little discussion of the social impact of this work. The new proportionality axioms (EJR, FJR) are described in theoretical detail, though not from a practical perspective. (This seems to be the norm for most social choice and mechanism design literature.)

The authors certainly could have done more to discuss the practical pros and cons of their axioms and Rule X, perhaps with the discussion of some experiments using real data. Their one example of Circleville (Fig. 1) is nice, but only one example. In particular, it would be helpful if the authors expand on how the practical implications of EJR/FJR differ from those of the core (or other notions of fairness and efficiency in PB).


**Main Review:**


This paper presents some interesting and novel theoretical results in participatory budgeting. The exposition and notation is very clear, and authors take care to place their contributions within the existing literature. The introduction of EJR and FJR, and their adaptation of Rule X to a general PB setting, are both notable contributions.

The proof of their main theoretical results (Theorem 1 and Example 1) are easy to follow, and I appreciate that these are explicitly included included in the main paper. Some of the other theoretical contributions are less impactful, as they are left to the appendix (which I did not read). I am somewhat familiar with this literature -- and by no means an expert -- and these results look sound to me.

One negative point: there are several experiments with real PB data mentioned in the conclusion, which are not discussed or presented in the main paper. The extent of these experiments could almost constitute a separate paper, and they don't seem to add much to this paper since they are only mentioned in passing. As an applied researcher, I would have enjoyed reading about these experiments, and the implications of this theoretical work on real PB elections. However these experiments are not very important to the main (theoretical) contributions of this paper, so the overall paper is not diminished by their absence.

**Time Spent Reviewing:**

1

---

> ### Author Response · Authors · 2021-08-09
> **Response to the reviewer's questions**
>
> Thanks a lot for the feedback! You’re right that we’ve focussed on theoretical results, but that more reflection on the social impact is in order. We have tightened the paper somewhat, and will use some of the extra space to extend our overview of the experiments in the appendix.
>
> You make an interesting suggestion of using experiments with real data to distinguish various fairness properties (like EJR and FJR) by their practical impact. We’ve run a preliminary experiment on datasets available from Pabulib (mostly from Poland), and in particular we checked whether the outcome that was in fact selected by the cities is fair according to the kind of fairness criteria we have been studying. We found that in at least 59 out of 366 elections (16%), EJR was failed. In most cases, the failure was of the form that there was a group of voters who approved 0 of the selected projects but who approved an unelected project in common, and the group was large enough to afford that project. In the committee context, this would be called a failure of JR. But we did find some examples, where JR was satisfied but EJR was not, so EJR provides additional strength. Unfortunately it is computationally quite expensive to search for these violations, so a thorough experiment will need to be left for future work.
>
> In the revised version, we will expand our discussion of the practical implications of the fairness axioms along the lines mentioned above. Of course, our existing experiments will be available in an online appendix.

---

### Official Review · Reviewer_yC2K · 2021-07-17

**Rating:** 6
**Confidence:** 4

**Summary:**

This paper studies the proportional participatory budgeting model, where the projects have arbitrary costs and the voters have additive utilities. In this classic model of PB, the paper studies fairness concept and explore whether existing rules satisfy certain fairness guarantee. Most of the results in this paper are based on approval-based committee elections and some concepts are slightly modified to fit the PB model.




**Limitations And Societal Impact:**

Yes

**Main Review:**

The major contributions are:

1. The paper extends Rule X to proportional participatory budgeting and the axiom named Extended Justified Representation (EJR), which is well-known in committee elections. It is easy to follow since they define it step by step by using several `intermediate’ definitions. They also show that Rule X satisfies EJR up to one project, a mild relaxation of EJR, and give a counterexample to show the PAV rule does not satisfy EJR. In addition, they show that Rule X satisfies the alpha-core with asymptotically tight bounds with respect to alpha and other properties like exhaustiveness.

2. The paper proposes a strengthening concept of EJR, namely FJR, which guarantees representation to groups that are only partially cohesive. Although Rule X fails FJR, they also give a new rule named GCR, which satisfies FJR.

3. The paper also do experiments comparing different variants of Rule X, Phragmén’s rule, and PAV and explore more about GCR.

Here are some concerns about the paper.

1. It looks like EJR and EJR up to one project is different in the general PB model. The authors only show that Rule X satisfies EJR up to one project, and PAV fails EJR. Does PAV fail EJR up to one project in the general PB model? It would be better to compare two rules with the same axiom in the general PB model. Is there any brief thinking about Rule X and EJR?

2.It would be better to give the definition consistently, like definition 7 uses |S|>cost(T)*n but definition 8 uses \sum_{c\in T}cost(c)\le |S|/n, both having the same meaning, but the former looks concise.

3. The authors also propose FJR. Are there any scenarios where the rules that satisfy EJR are not good, so we need to use rules that satisfy FJR?

The authors addressed my questions.

**Time Spent Reviewing:**

15

---

> ### Author Response · Authors · 2021-08-09
> **Response to the reviewer's questions**
>
> Thanks a lot for the feedback!
>
> 1. Regarding EJR and EJR1 (EJR up to one project): For PAV, Example 1 shows that PAV fails EJR1. In fact it shows a much stronger statement: that for each constant $r > 1$ PAV fails EJR up-to-$r$. We’ll say this explicitly. For Rule X, in the submission we said “We do not know if Rule X satisfies EJR in the general PB model” (l. 180). We have since settled this: Rule X does not satisfy EJR, so the relaxation to EJR1 is needed. There is a small example which we will include in the paper, but here is another nice way to see this: For the case of $n=1$ (a single voter), we are looking at the standard knapsack problem, and EJR is strong enough to require that we have to choose the optimum knapsack (with respect to the single voter’s utility function). But Rule X is strongly polynomial time, so it cannot possibly satisfy EJR unless P=NP.
>
> 2. Thanks. We will reformulate the condition in Definition 8 as you suggest to make it consistent.
>
> 3. “Are there any scenarios where the rules that satisfy EJR are not good, so we need to use rules that satisfy FJR?”: Our motivation for strengthening EJR to FJR is to give guarantees to groups of voters even if they aren’t unanimous in their objections but still have enough “common ground”. For committee elections, PAV satisfies EJR but not the core, and notably the examples commonly presented where PAV fails core are actually examples where PAV fails FJR (e.g., Peters and Skowron 2020 or Cheng et al 2019), and these papers criticizing PAV provide motivation for looking for FJR rules. Whether this makes a difference in practical data sets is an important question, but (in our experience) it seems computationally very difficult to search for FJR violations using for example an ILP solver, so more work is required to illuminate this issue.

---

### Official Review · Reviewer_tQCy · 2021-07-19

**Rating:** 6
**Confidence:** 3

**Summary:**

The paper deals with a setting of participatory budgeting in which voters's preferences over projects are aggregated to fund a subset of the candidate projects with a limited budget. Based on justified representation in committee elections (a related setting without costs) a fairness concept is introduced as well as an aggregation rule (named Rule X) which is shown to satisfy the concept, while other commonly used rules fail to do so. Several variants, restrictions and extensions are discussed.

**Ethical Concerns:**

-

**Limitations And Societal Impact:**

-

**Main Review:**

The topic of participatory budgeting and the suggested model are, as far as I know, in the scope of Neurips. The results and open questions significant and original as far as I can tell. The models and their theoretical analysis appear reasonable to me.

To my mind, related work should be elaborated on in the introduction. Some information is only provided later (see for example Section 3.1, middle and bottom of page 4), and should be mentioned early on. For instance, a reference for the concept justified respresentation is missing in the introduction (cf p. 1 and p. 3). "Many works" (l. 39) are referred to without literature pointers. The introductory example uses some general statements without reference such has "most cities" (l. 43), "many cities" (l. 55).

I like the readability of the problem setting and the main part of the paper. Especially, the intuition and formality, and the comparison to the state of the art in committee elections and a permanent review of where the results can be settled in comparison was quite helpful. At some points defensive statements like "our proofs are significantly different from..." and "This is yet another explanation why EJR is am important and well-justified axiom" seemed a little misplaced and too meta-explanatory to me.

Considering Example 1 (p. 7), I get the impression that the condition is too strong to hold for a single voter. Even if that player forms a T-cohesive subset, it might make sense to relax the EJR property to large enough groups. What happens to rules like PAV for such a variant?

Proofs and additional details are often deferred to the appendix. From my point of view, the paper should be self-contained. Hence, it would be better to include at least sketches in the main paper and omit references to an appendix.
Similarly, Section 3.3 and the conclusion should be rewritten to present information differently.
Without the appendix many details are not comprehensible. For instance, I have not verified the proofs of Theorem 2 (Appendix D) and Theorem 3 (Appendix F). To my mind, the latter analysis of a new rule satisfying FJR is one of the main claims in the paper and should have more focus. Statements such as "in Appendix F we discuss other important properties as well as some drawbacks of GCR" (l.328) do not provide much valuable information in the main paper.

At the same time Figure 1 takes a lot of space for example details that are barely used. The picture is only referred to in one paragraph, and many variables are not even used.

The example of a "bike trail along Example River" (l. 66) additionally involves the issue of dependencies of projects which is not studied here. I wonder what can be said about such additional constraints.


Minor details:

l. 49: "so we can assume..." the following statement might work for the example but this is assumption cannot be implied (without reference from social sciences). It would be better to rephrase this less generalized.

l. 102: Please rephrase "that is"; what follows is only an implication.

l. 110: A Committee size |W| can also be less than k?

p.4: The notion mathcal{R}(E) is not defined before Definition 2, and can easily be added in the preliminaries.

p. 8: 3.3: Priceability is not defined at all.
      4.1: The tie breaking procedure is mentioned to be needed in the context of priceability, which is not studied in the main paper. For the sake of self-containment, perhaps the information "breaking ties in favor of smaller cost(T)" can be written in Footnote 6, if not left out completely

**Time Spent Reviewing:**

20

---

> ### Author Response · Authors · 2021-08-09
> **Response to the reviewer's questions**
>
> Thanks for your helpful review!
>
> We appreciate your comments regarding what to defer to the appendix. We have experimented with what’s possible within the page limit, and by squeezing Figure 1 and tightening the discussion of the core, we are able to include the full proof of Theorem 3 and more details of properties of GCR (from Appendix F). We will also use this space for a better handling of experiments in our discussion section. We also plan to make the appendix with all proof details publicly available.
>
> Regarding Example 1, and whether PAV satisfies EJR for large groups: There are related examples showing that PAV fails EJR (and even relaxed variants) also for larger groups. Of course, we could fix a natural number $z$ and clone each voter $z$ times. Then, the result returned by PAV will not change since the PAV score of each committee will increase by a multiplicative factor of $z$. This shows that PAV fails the axiom for groups containing multiple voters. One can also ask whether PAV satisfies EJR up-to-one for groups which are large enough relative to the size of the whole population of voters. Example 1 with $r=2$ shows that PAV fails the axiom even for groups that contain 25% of the voters. This example can be further tweaked to show that for each $\alpha < 1$ PAV may fail the axiom even for groups of size $\alpha$ times the number of voters:
>
> Let $r \geq 2$.
>
> 1 voter approving $A_1, A_2, …, A_{r}$
>
> $r-1$ voters approving  $B_1, …, B_{r^3 + r}$
>
> Assume $A_i$ costs $1/r$ and $B_i$ costs $1/r^4$. Then the cohesive group of $(r-1)$ voters deserves all $r^3 + r$ candidates. PAV will select only $(r-1)$ $A$-candidates and $r^3$ $B$-candidates.
>
> At the same time, Rule X provides guarantees even for single voters.

---

### Decision · Program_Chairs · 2021-09-27

**Decision:**

Accept (Poster)

**Comment:**

This is probably the strongest paper in my batch (as AC).